# Calcium dependence of neurotransmitter release at a high fidelity synapse

Abdelmoneim Eshra, Hartmut Schmidt, Jens Eilers, Stefan Hallermann*

Carl-Ludwig-Institute for Physiology, Medical Faculty, University of Leipzig, Leipzig, Germany

**Abstract** The $Ca^{2+}$-dependence of the priming, fusion, and replenishment of synaptic vesicles are fundamental parameters controlling neurotransmitter release and synaptic plasticity. Despite intense efforts, these important steps in the synaptic vesicles' cycle remain poorly understood due to the technical challenge in disentangling vesicle priming, fusion, and replenishment. Here, we investigated the $Ca^{2+}$-sensitivity of these steps at mossy fiber synapses in the rodent cerebellum, which are characterized by fast vesicle replenishment mediating high-frequency signaling. We found that the basal free $Ca^{2+}$ concentration (<200 nM) critically controls action potential-evoked release, indicating a high-affinity $Ca^{2+}$ sensor for vesicle priming. $Ca^{2+}$ uncaging experiments revealed a surprisingly shallow and non-saturating relationship between release rate and intracellular $Ca^{2+}$ concentration up to 50 µM. The rate of vesicle replenishment during sustained elevated intracellular $Ca^{2+}$ concentration exhibited little $Ca^{2+}$-dependence. Finally, quantitative mechanistic release schemes with five $Ca^{2+}$ binding steps incorporating rapid vesicle replenishment via parallel or sequential vesicle pools could explain our data. We thus show that co-existing high- and low-affinity $Ca^{2+}$ sensors mediate priming, fusion, and replenishment of synaptic vesicles at a high-fidelity synapse.

*For correspondence:
stefan_jens.hallermann@uni-leipzig.de

Competing interests: The authors declare that no competing interests exist.

## Introduction

Neurotransmitter release is mediated by the presynaptic vesicle cycle (*Südhof, 2004*) including (1) the priming of neurotransmitter-filled vesicles, (2) the fusion of primed vesicles, and (3) the replenishment of new vesicles after fusion. The $Ca^{2+}$-sensitivity of these steps is difficult to determine due to the large spatial gradients of the $Ca^{2+}$ concentration, which occur during $Ca^{2+}$ influx through the $Ca^{2+}$ channels. While the basal free intracellular $Ca^{2+}$ concentration is ~50 nM, thousandfold higher local microdomains of $Ca^{2+}$ build and decay very fast around the $Ca^{2+}$ channels (*Simon and Llinás, 1985*; *Yamada and Zucker, 1992*). The small size and the rapid kinetics of the microdomain signals complicate the quantification of the local $Ca^{2+}$ signals with the imaging techniques (*Neher, 1998*). $Ca^{2+}$ uncaging circumvented this problem by allowing for the homogenous elevation of $Ca^{2+}$ concentration throughout the whole presynaptic compartment via UV-photolysis of caged $Ca^{2+}$ compounds (*Kaplan and Ellis-Davies, 1988*) and thus the direct measurement of the $Ca^{2+}$-concentration immediately relevant for vesicle fusion (reviewed by *Neher, 1998*; *Kochubey et al., 2011*).

Among the steps of the presynaptic vesicle cycle, the $Ca^{2+}$-sensitivity of vesicle fusion is best studied. First experiments with $Ca^{2+}$ uncaging at retinal bipolar cells of goldfish found a very low sensitivity of the release sensors with a half saturation at ~100 µM $Ca^{2+}$ concentration and a fourth to fifth power relationship between $Ca^{2+}$ concentration and neurotransmitter release (*Heidelberger et al., 1994*), similar to previous estimates at the squid giant synapse (*Adler et al., 1991*; *Llinás et al., 1992*). Subsequent work at other preparations showed different dose-response curves. For example, analysis of a central excitatory synapse, the calyx of Held (*Forsythe, 1994*) at a young pre-hearing age, found a much higher affinity with significant release below 5 µM intracellular $Ca^{2+}$ concentration and a steep dose-response curve (*Bollmann et al., 2000*; *Lou et al., 2005*;

*Schneggenburger and Neher, 2000*; *Sun et al., 2007*). Further analysis of the calyx of Held during neuronal development comparing the $Ca^{2+}$-sensitivity of the release sensors at the age of P9 to P12-P15 (*Kochubey et al., 2009*) and P9 to P16-P19 (*Wang et al., 2008*) showed a developmental decrease in the $Ca^{2+}$-sensitivity of vesicle fusion. Studies at two other central synapses, the hippocampal mossy fiber boutons of rats (P18–30; *Fukaya et al., 2021*) and the boutons of cerebellar basket cells of mice (P11-16; *Sakaba, 2008*), also described a high $Ca^{2+}$-sensitivity of vesicle fusion with a steep dose-response curve. In contrast, the dose-response curve of sensory neurons of the rod photoreceptors was more shallow (*Duncan et al., 2010*; *Thoreson et al., 2004*) and vesicle fusion below 7 µM $Ca^{2+}$ concentration was absent at the cochlear inner hair cells (*Beutner et al., 2001*).

The steps preceding the fusion of synaptic vesicles are in general still poorly understood (*Südhof, 2013*). We refer to vesicle priming as the molecular and positional preparation of vesicles for fusion near $Ca^{2+}$ channels (*Neher and Sakaba, 2008*). Molecular priming has recently been shown to be the functional correlate of vesicle docking (*Imig et al., 2014*; *Maus et al., 2020*). Vesicle replenishment refers to the delivery of new vesicles during sustained activity. The effect of the residual $Ca^{2+}$ on the strength of synapses particularly during synaptic facilitation has been studied for decades with a particular focus on the release probability of vesicles (see Discussion). Here, we investigate the $Ca^{2+}$-dependence of priming and replenishment, which increases the number of release-ready vesicles. Previous work provided evidence that priming and replenishment are strongly $Ca^{2+}$-dependent (reviewed by *Silva et al., 2021*, and *Neher and Sakaba, 2008*). The following findings demostrate the $Ca^{2+}$-dependence of vesicle priming and replenishment. First, the size of the pool of fast-releasing vesicles linearly depends on the intracellular $Ca^{2+}$ concentration at the calyx of Held synapse (*Hosoi et al., 2007*; see also *Awatramani et al., 2005*; *Wang and Kaczmarek, 1998*). Second, the sustained component of release, presumably reflecting vesicle replenishment, linearly depends on the intracellular $Ca^{2+}$ concentration at cerebellar basket cell synapses (*Sakaba, 2008*). Third, the number of docked vesicles assessed by electron microscopic techniques is rapidly and reversibly regulated depending on the resting $Ca^{2+}$ levels and neuronal activity at hippocampal neurons (*Chang et al., 2018*; *Imig et al., 2020*; *Kusick et al., 2020*; *Vandael et al., 2020*; *Vevea et al., 2021*). Fourth, the occupancy of the docking sites increases upon elevating extracellular $Ca^{2+}$ levels at cerebellar synapses (*Blanchard et al., 2020*; *Malagon et al., 2020*). Finally, in several studies on chromaffin cells and synapses of vertebrates and invertebrates, the assumption of $Ca^{2+}$-dependent priming was required to explain the experimental data (*Doussau et al., 2017*; *Kobbersmed et al., 2020*; *Millar et al., 2005*; *Pan and Zucker, 2009*; *Voets, 2000*; *Walter et al., 2013*). In contrast, previous studies at cerebellar mossy fiber synapses could explain release during trains of action potentials or prolonged depolarizations with $Ca^{2+}$-independent vesicle priming and replenishment (*Hallermann et al., 2010*; *Ritzau-Jost et al., 2014*; *Ritzau-Jost et al., 2018*; *Saviane and Silver, 2006*).

The discrepant findings of the $Ca^{2+}$-sensitivity of vesicle priming, fusion, and replenishment could be due to methodological errors. However, synapses show type-specific functional and structural differences (*Atwood and Karunanithi, 2002*; *Nusser, 2018*; *Zhai and Bellen, 2004*). The rate at which vesicles are replenished to empty release sites seems to be particularly different between types of synapses. The cerebellar mossy fiber bouton (cMFB) conveys high-frequency sensory information to the cerebellar cortex and relies on extremely fast vesicle replenishment (*Miki et al., 2020*; *Ritzau-Jost et al., 2014*; *Saviane and Silver, 2006*). The aim of this study was therefore to determine the $Ca^{2+}$-sensitivity of vesicle priming, fusion, and replenishment at mature cMFBs synapses at physiological temperature, and to test whether and how the prominent fast vesicle replenishment affects the $Ca^{2+}$-dependence of the vesicle priming, fusion, and replenishment at this synapse. To measure the $Ca^{2+}$-dependence of vesicle priming, we first directly manipulated the free basal intracellular $Ca^{2+}$ concentration and measured the amount of action potential-evoked release. To meaure the $Ca^{2+}$-dependence of vesicle fusion, we focused the initial release kinetics of the fusion of the primed vesicles upon $Ca^{2+}$ uncaging (with time constants mostly << 10 ms). To finally measure the $Ca^{2+}$-dependence of vesicle replenishment, we focused on the sustained component of release occurring during 100 ms of flash-evoked $Ca^{2+}$ increase.

Our data revealed a strong dependence of the number of release-ready vesicles on basal $Ca^{2+}$ concentrations between 30 and 180 nM, a significant release below 5 µM, an apparent shallow dose-response curve in the studied $Ca^{2+}$ concentration range of 1 to 50 µM, and little $Ca^{2+}$-dependence of vesicle replenishment during sustained elevated intracellular $Ca^{2+}$ concentrations.

Computational simulations incorporating mechanistic release schemes with five $Ca^{2+}$ binding steps and fast vesicle replenishment via sequential or parallel pools of vesicles could explain our data. Our results show the co-existence of $Ca^{2+}$ sensors with high- and low-affinities that cover a large range of intracellular $Ca^{2+}$ concentrations and mediate fast signaling at this synapse.

## Results

### Action potential-evoked synaptic release critically depends on basal intracellular $Ca^{2+}$ concentration

To investigate the impact of the basal intracellular $Ca^{2+}$ concentration on synaptic release, we performed simultaneous patch-clamp recordings from presynaptic cerebellar mossy fiber boutons (cMFB) and postsynaptic granule cells (GC) of 5- to 6-week-old mice at physiological temperatures (*Figure 1A and B*). We aimed at clamping the free $Ca^{2+}$ concentration in the presynaptic patch solution to either low or high basal $Ca^{2+}$ concentrations by adding different concentrations of $Ca^{2+}$ and the $Ca^{2+}$ chelator EGTA (see Materials and methods). Two-photon quantitative $Ca^{2+}$ imaging with the dual-indicator method using Fluo-5F as the $Ca^{2+}$ indicator (*Delvendahl et al., 2015*; *Sabatini et al., 2002*) revealed the free $Ca^{2+}$ concentration of the presynaptic intracellular solution to be 28 ± 3 and 183 ± 8 nM, for the low and high basal $Ca^{2+}$ conditions (n = 4 and 4), respectively (*Figure 1A*). In both solutions, the free EGTA concentration was 4.47 mM (see Materials and methods). In response to triggering a single action potential in the presynaptic

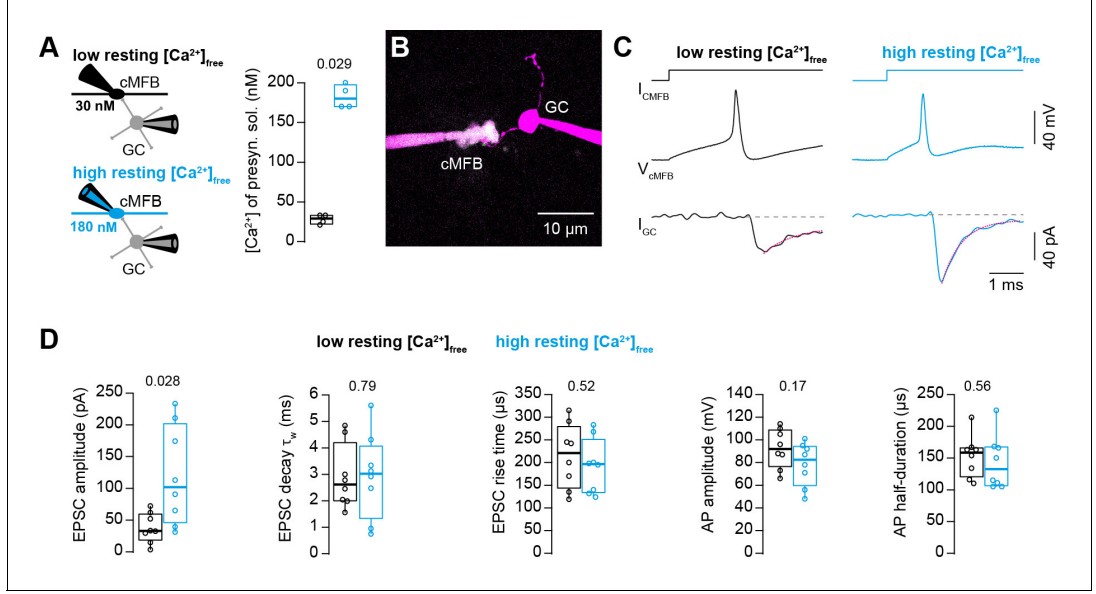

**Figure 1.** Action potential-evoked synaptic release critically depends on basal intracellular $Ca^{2+}$ concentration. (**A**) *Left:* Illustration of the cellular connectivity of the cMFB to GC synapse during simultaneous pre- and postsynaptic patch-clamp recording. The presynaptic terminal was loaded with an intracellular solution having either low or high free basal $Ca^{2+}$ concentration (top and bottom, respectively). *Right:* Comparison of the average free $Ca^{2+}$ concentration in the presynaptic patch pipette (quantified by two-photon $Ca^{2+}$ imaging) for the intracellular solutions with low and high basal $Ca^{2+}$ (n = 4 each). (**B**) Example two-photon microscopic image of a cMFB and a GC in the paired whole-cell configuration. (**C**) Example traces of a paired cMFB-GC recording with current injection ($I_{cMFB}$) (*top*) eliciting an action potential in the cMFB (*middle*) and an EPSC in the postsynaptic GC (*bottom*). Black and blue color code corresponds to low and high free basal $Ca^{2+}$ concentration in the presynaptic solution, respectively. The decay of the EPSC was fitted with a bi-exponential function (magenta line). (**D**) Comparison of the properties of presynaptic action potentials and EPSCs evoked after eliciting an action potential in the presynaptic terminal using solutions having either low (black) or high (blue) free $Ca^{2+}$ concentration. From left to right: peak amplitude of the EPSC, weighted decay time constant of the EPSC, 10-to-90% rise time of the EPSC, amplitude of the presynaptic action potential, and action potential half-duration (n = 8 and 8 paired cells for the conditions with low and high resting $Ca^{2+}$ concentration, respectively). Boxplots show median and 1st/3rd quartiles with whiskers indicating the whole data range. Values of individual experiments are superimposed as circles. The numbers above the boxplots represent p-values of Mann-Whitney *U* tests.

The online version of this article includes the following source data for figure 1:

**Source data 1.** Action potential-evoked synaptic release critically depends on basal intracellular $Ca^{2+}$ concentration.

terminal, the recorded excitatory postsynaptic current (EPSC) depended strongly on the presynaptic resting $Ca^{2+}$ concentration (*Figure 1C*). We found an almost threefold increase in the EPSC amplitude when elevating the resting $Ca^{2+}$ concentration in the presynaptic terminals from 30 to 180 nM. On average, the EPSC amplitudes were $39 \pm 8$ and $117 \pm 28$ pA for the low and high basal $Ca^{2+}$ conditions, respectively (n = 8 and 8; $P_{Mann-Whitney}$ = 0.028; *Figure 1D*). Interestingly, the frequency of miniature currents in-between the current injections used to elicit action potentials had a tendency to increase with elevated basal $Ca^{2+}$ concentration (median 1.1 and 3.5 Hz for the low and high basal $Ca^{2+}$ conditions, respectively, n = 8 and 8; $P_{Mann-Whitney}$ = 0.13; data not shown). The EPSC rise and decay kinetics were not significantly different (*Figure 1D*). No significant differences were observed in the action potential waveform including amplitude and half duration (*Figure 1D*) indicating that the altered synaptic strength was not caused by changes in the shape of the presynaptic action potential. These data indicate that moderate changes in the presynaptic basal $Ca^{2+}$ concentration can alter synaptic strength up to threefold.

## $Ca^{2+}$ uncaging dose-response curve measured with presynaptic capacitance measurements

To gain a better understanding of the profound sensitivity of AP-evoked release on presynaptic basal $Ca^{2+}$ concentration, we established presynaptic $Ca^{2+}$ uncaging and measured the release kinetics upon step-wise elevation of $Ca^{2+}$ concentration. We combined wide-field illumination using a high-power UV laser with previously established quantitative two-photon $Ca^{2+}$ imaging (*Delvendahl et al., 2015*) to quantify the post-flash $Ca^{2+}$ concentration (*Figure 2A*). This approach offers sub-millisecond control of the UV flashes and a high signal to noise ratio of the two-photon $Ca^{2+}$ imaging deep within the brain slice. The flash-evoked artefacts in the two-photon signals, presumably due to luminescence in the light path, could be reduced to a minimum with an optimal set of spectral filters and gate-able photomultipliers (PMTs). Subtraction of the remaining artefact in the background region of the two-photon line scan resulted in artefact-free fluorescence signals (*Figure 2B and C*).

To obtain a large range of post-flash $Ca^{2+}$ concentrations within the bouton, we varied the concentration of the $Ca^{2+}$-cage DMn (1–10 mM) and the intensity (10–100%) and the duration (100 or 200 µs) of the UV laser pulse (*Table 1*). The spatial homogeneity of the $Ca^{2+}$ elevation was assessed by UV illumination of caged fluorescein mixed with glycerol (*Figure 2—figure supplement 1*; *Schneggenburger and Neher, 2000*; *Bollmann et al., 2000*). The resulting post-flash $Ca^{2+}$ concentration was quantified with either high- or low-affinity $Ca^{2+}$ indicator (Fluo-5F or OGB-5N). To measure the kinetics of neurotransmitter release independent of dendritic filtering or postsynaptic receptor saturation, vesicular fusion was quantified by measuring the presynaptic capacitance with a 5 kHz-sinusoidal stimulation (*Hallermann et al., 2003*). The first 10 ms of the flash-evoked capacitance increase was fitted with functions containing a baseline and mono- or bi-exponential components (magenta line in *Figure 2D and E*; see *Equation 1* in the Materials and methods section). With increasing post-flash $Ca^{2+}$ concentration the fast time constant decreased ($\tau$ in case of mono- and $\tau_1$ in case of bi-exponential fits; *Figure 2D*). The inverse of the fast time constant represents a direct readout of the fusion kinetics of the release-ready vesicles. When plotting the inverse of the time constant as a function of post-flash $Ca^{2+}$ concentration, we obtained a shallow dose-response curve that showed a continuous increase in the release rate with increasing post-flash $Ca^{2+}$ concentration up to 50 µM (*Figure 2F*). In some experiments with high $Ca^{2+}$ concentrations, the release was too fast to be resolved with 5 kHz capacitance sampling (i.e. time constants were smaller than 200 µs; *Figure 2E*). We therefore increased the frequency of the sinusoidal stimulation in a subset of experiments to 10 kHz (15 out of 80 experiments). Such high-frequency capacitance sampling is to our knowledge unprecedented at central synapses and technically challenging because exceptionally low access resistances are required (<~15 MΩ) to obtain an acceptable signal-to-noise ratio (*Gillis, 1995*; *Hallermann et al., 2003*). Despite these efforts, the time constants were sometimes faster than 100 µs, representing the resolution limit of 10 kHz capacitance sampling (*Figure 2E*). These results indicate that the entire pool of release-ready vesicles can fuse within less than 100 µs. Fitting a Hill equation on both 5- and 10 kHz data resulted in a best-fit $K_D$ of >50 µM with a best-fit Hill coefficient, *n*, of 1.2 (*Figure 2F*).

In addition to the speed of vesicle fusion, we analyzed the delay from the onset of the UV-illumination to the onset of the rise of membrane capacitance, which was a free parameter in our fitting

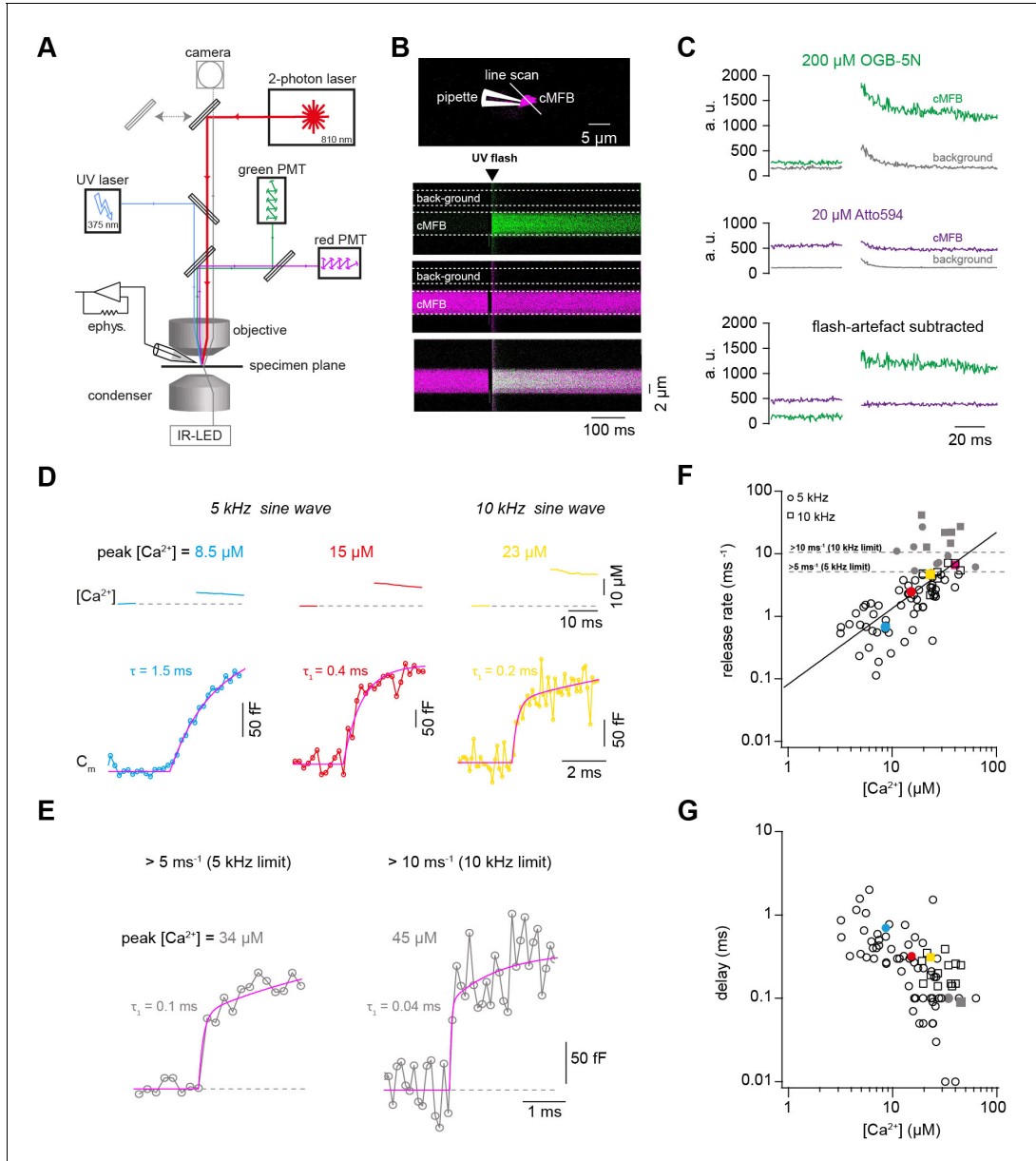

**Figure 2.** Ca$^{2+}$ uncaging dose-response curve measured with presynaptic capacitance measurements. (**A**) Illustration of the experimental setup showing the light path of the two-photon laser illumination (red line), the UV laser illumination (blue line), the electrophysiology amplifier ('ephys.'), the red and green gate-able photomultiplier tubes (PMTs), and infrared LED illumination with oblique illumination via the condenser for visualization of the cells at the specimen plane by the camera (gray line) when the upper mirror is moved out of the light path (gray arrow). (**B**) *Top:* Two-photon microscopic image of a cMFB in the whole-cell configuration loaded with OGB-5N, Atto594, and DMn/ Ca$^{2+}$. Positions of the patch pipette and line scan are indicated. *Bottom:* Two-photon line scan showing the fluorescence signal as measured through the green PMT, red PMT, and an overlay of the green and red channels. Arrow indicates the onset of the UV flash and dashed lines represent the flash-induced luminescence artefact as detected outside the cMFB. The lookup tables for the green and red channel were arbitrarily adjusted independent of the absolute values in C. (**C**) *Top:* change in fluorescence intensity within the cMFB for the green channel along with the corresponding flash-induced green artefact measured in the background. *Middle:* change in fluorescence intensity within the cMFB for the red channel along with the corresponding flash-induced red artefact. *Bottom:* green and red fluorescence signal after subtracting the flash-induced artefacts. (**D**) *Top:* Ca$^{2+}$ signals of different concentrations elicited through Ca$^{2+}$ uncaging in three different cells, the flash was blanked. *Bottom:* corresponding traces of capacitance recordings measured using a 5 kHz (left and middle) or 10 kHz sinusoidal stimulation (right). $\tau$ represents the time constant from a mono-exponential fit, $\tau_1$ represents the time constant of the fast component of a bi-exponential fit. (**E**) Traces of capacitance recordings showing the resolution limit in detecting fast release rates of >5 ms$^{-1}$ using 5 kHz sinusoidal stimulation or >10 ms$^{-1}$ using 10 kHz sinusoidal stimulation. (**F**) Plot of release rate versus post-flash Ca$^{2+}$ concentration (n = 65 from 5-kHz- and from 15 10-kHz-recordings obtained from 80 cMFBs). The line represents a fit with a Hill equation (*Equation 2*) with best-fit values $V_{max}$ = 1.7*10$^7$ ms$^{-1}$, $K_D$ = 7.2*10$^6$ μM, and *n* = 1.2. Color coded symbols correspond to traces in (**D** – **E**). Gray symbols represent values above the resolution

*Figure 2 continued on next page*

*Figure 2 continued*

limit. (**G**) Plot of synaptic delay versus post-flash Ca$^{2+}$ concentration (n = 64 from 5-kHz- and 15 from 10-kHz-recordings obtained from 79 cells). Note that one recording was removed from the analysis because the exponential fit led to a negative value of the delay. Color coded symbols correspond to traces in (**D** – **E**).

The online version of this article includes the following source data and figure supplement(s) for figure 2:

**Source data 1.** Ca$^{2+}$ uncaging dose-response curve measured with presynaptic capacitance measurements.

**Figure supplement 1.** Measurement of the UV energy profile with caged fluorescein.

functions (see *Equation 1*). The delay was strongly dependent on the post-flash Ca$^{2+}$ concentration and the dose-response curve showed no signs of saturation at high Ca$^{2+}$ concentrations (*Figure 2G*), which is consistent with the non-saturating release rates. These data reveal that the fusion kinetics of synaptic vesicles increased up to a Ca$^{2+}$ concentration of 50 µM without signs of saturation, suggesting a surprisingly low apparent affinity of the fusion sensor at mature cMFBs under physiological temperature conditions ($K_D$ > 50 µM).

## Ca$^{2+}$ uncaging dose-response curve measured with deconvolution of EPSCs

Capacitance recordings are not very sensitive in detecting low release rates. We therefore performed simultaneous pre- and postsynaptic recordings and used established deconvolution techniques to calculate the presynaptic release rate by analyzing the EPSC as previously applied at this synapse (*Figure 3A,B*; *Ritzau-Jost et al., 2014*). Kynurenic acid (2 mM) and cyclothiazide (100 µM) were added to the extracellular solution in order to prevent the saturation and desensitization of postsynaptic AMPA receptors, respectively. Ca$^{2+}$ uncaging in the presynaptic terminal evoked EPSCs with kinetics, which strongly depended on the post-flash Ca$^{2+}$ concentration. The cumulative release obtained from deconvolution analysis of the recorded EPSCs was fitted as previously done for capacitance traces (*Equation 1*). At low Ca$^{2+}$ concentrations (<5 µM), a significant amount of neurotransmitter release could be measured, which is consistent with previous reports from central synapses (*Bollmann et al., 2000*; *Fukaya et al., 2021*; *Sakaba, 2008*; *Schneggenburger and Neher, 2000*). The presynaptic release rates increased with increasing post-flash Ca$^{2+}$ concentration and no saturation in the release rate occurred in the dose-response curve (*Figure 3D*). The dose-response curve for the delay from the onset of the UV illumination to the onset of the rise of the cumulative release trace (*Equation 1*) did not show signs of saturation of the release kinetics in the

**Table 1.** Parameters for weak, middle, and strong post-flash Ca$^{2+}$ elevations.

|  | weak Ca$^{2+}$ elevation | middle Ca$^{2+}$ elevation | strong Ca$^{2+}$ elevation |
|---|---|---|---|
| UV illumination |  |  |  |
| Duration (ms) | 0.1 or 1 | 0.1 | 0.1 or 0.2 |
| Intensity (%) | 10–100 | 20–100 | 100 |
| Concentration in intracellular solution (mM) |  |  |  |
| ATTO 594 | 0.010 | 0.020 | 0.020 |
| Fluo 5F | 0.050 | 0 | 0 |
| OGB 5N | 0 | 0.200 | 0.200 |
| CaCl2 | 0.500 | 2.000 | 10.000 |
| DM-N | 0.500 | 2.000 | 10.000 |
| Obtained peak post-flash Ca$^{2+}$ (µM) |  |  |  |
| Min | 1.1 | 2.7 | 15.7 |
| Max | 7.1 | 36.0 | 62.6 |
| Median | 2.4 | 8.8 | 25.1 |
| Simulated uncaging fraction of DMn |  |  |  |
| α | 0.08–0.5 | 0.15–0.55 | 0.14–0.25 |

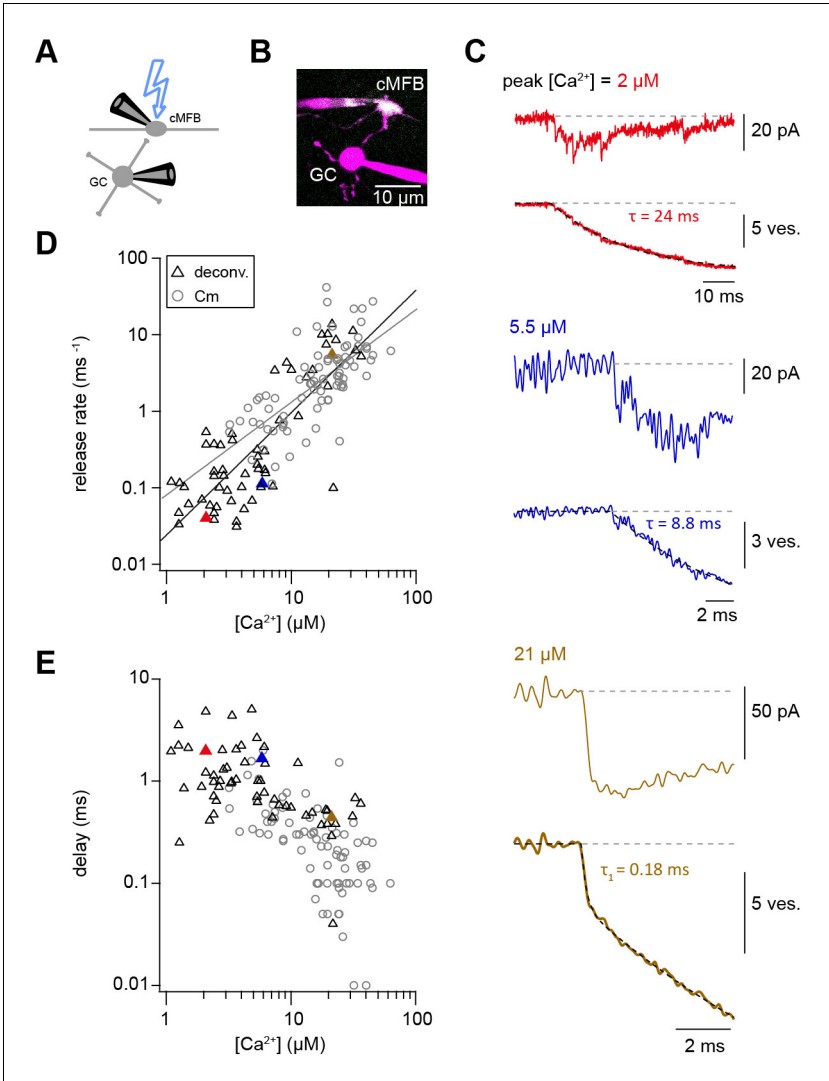

**Figure 3.** Ca$^{2+}$ uncaging dose-response curve measured with deconvolution of EPSCs. (**A**) Illustration of the cellular connectivity in the cerebellar cortex showing the pre- and postsynaptic compartments during paired whole-cell patch-clamp recordings and Ca$^{2+}$ uncaging with UV-illumination. (**B**) Two-photon microscopic image of a cMFB and a GC in the paired whole-cell patch-clamp configuration. (**C**) Three different recordings showing UV-flash evoked EPSC (*top trace*) and cumulative release rate measured by deconvolution analysis of the EPSCs (*bottom trace*). The peak Ca$^{2+}$ concentration, quantified with two-photon Ca$^{2+}$ imaging, is indicated in each panel. $\tau$ represents the time constant from mono-exponential fit, $\tau_1$ represents the time constant of the fast component of bi-exponential fit. Note the different lengths of the baselines in the three recordings. (**D**) Plot of release rate versus post-flash Ca$^{2+}$ concentration. Gray open circles represent data from capacitance measurements (*Figure 2*) and black triangles represent data from deconvolution analysis of EPSC (n = 57 recordings obtained from 42 paired cells). Gray and black lines represent fits with a Hill equation of the capacitance (as shown in *Figure 1F*) and the deconvolution data, respectively. The best-fit parameters for the fit on the deconvolution data were $V_{max}$ = 6*10$^7$ ms$^{-1}$, $K_D$ = 7.6*10$^5$ µM, and n = 1.6. Red, blue, and brown symbols correspond to the traces in (**C**). (**E**) Plot of synaptic delay versus post-flash Ca$^{2+}$ concentration (n = 59 recordings obtained from 43 paired cells). Note that two recordings was removed from the analysis because the exponential fit led to a negative value of the delay Gray open circles represent data from capacitance measurements, and black triangles represent data from deconvolution analysis of EPSC. Red, blue, and brown symbols correspond to the traces in (**C**).

The online version of this article includes the following source data and figure supplement(s) for figure 3:

**Source data 1.** Ca$^{2+}$ uncaging dose-response curve measured with deconvolution of EPSCs.

**Figure supplement 1.** Measuring the $K_D$ of the Ca$^{2+}$ sensitive dyes.

**Figure supplement 2.** Comparison of brief versus long UV illumination to rule out fast Ca$^{2+}$ overshoots.

*Figure 3 continued on next page*

*Figure 3 continued*

**Figure supplement 3.** Correction for the post-flash changes in the fluorescent properties of the intracellular solution.

**Figure supplement 4.** Comparison of the time constants obtained from presynaptic capacitance measurements ($\tau_{Cm}$) and analysis of postsynaptic current recordings ($\tau_{deconv}$).

investigated range. Thus, consistent with capacitance measurements, deconvolution analysis of post-synaptic currents revealed a shallow $Ca^{2+}$-dependence of neurotransmitter release kinetics (*Figure 3D and E*). Fitting a Hill equation to the deconvolution data resulted in a best-fit $K_D$ >50 µM and a Hill coefficient of 1.6 (*Figure 3D*). Therefore, two independent measures of synaptic release (presynaptic capacitance measurements and postsynaptic deconvolution analysis) indicate a non-saturating shallow dose-response curve up to ~50 µM.

To rule out methodical errors that might influence the dose-response curve, we carefully determined the $K_D$ of the $Ca^{2+}$ indicator OGB-5N using several independent approaches including direct potentiometry (*Figure 3—figure supplement 1*), because this value influences the estimate of the $Ca^{2+}$ affinity of the fusion sensors linearly. We estimated a $K_D$ of OGB-5N of ~30 µM being at the lower range of previous estimates ranging from 20 to 180 µM (*Delvendahl et al., 2015*; *DiGregorio and Vergara, 1997*; *Neef et al., 2018*), arguing against an erroneously high $K_D$ of the $Ca^{2+}$ indicator as a cause for the non-saturation.

In addition, we used the two following independent approaches to rule out a previously described $Ca^{2+}$ overshoot immediately following the UV illumination. Such a $Ca^{2+}$ overshoot would be too fast to be detected by the $Ca^{2+}$ indicators (*Bollmann et al., 2000*) but could trigger strong release with weak UV illumination which would predict a shallow dose-response curve. First, the time course of $Ca^{2+}$ release from DMn was simulated (see below; Figure 6A) and no significant overshoots were observed (see below). Secondly, we experimentally compared strong and short UV illumination (100% intensity; 0.1 ms) with weak and long UV illumination (10% intensity; 1 ms), because a $Ca^{2+}$ overshoot is expected to primarily occur with strong and short UV illumination. Comparison of these two groups of UV illumination resulted in similar post-flash concentrations but did not reveal a significant difference in the corresponding release rate indicating that undetectable $Ca^{2+}$ overshoots did not affect the measured release rate (*Figure 3—figure supplement 2*). Therefore, both approaches argue against a $Ca^{2+}$ overshoot as an explanation for the shallow dose-response curve.

## Presynaptic and postsynaptic measurements reveal two kinetic processes of neurotransmitter release

In some $Ca^{2+}$ uncaging experiments, synaptic release appeared to have two components, which could be due to heterogeneity amongst release-ready vesicles. We therefore systematically compared mono- and bi-exponential fits to the capacitance and deconvolution data (*Figure 4A B*). Several criteria were used to justify a bi-exponential fit (see Materials and methods). One criterion was at least a 4% increase in the quality of bi- compared with mono-exponential fits as measured by the sum of squared differences between the fit and the experimental data ($\chi^2$; *Figure 4D*). Consistent with a visual impression, this standardized procedure resulted in the classification of ~40% of all recordings as bi-exponential (38 out of 80 capacitance measurements and 17 out of 59 deconvolution experiments; *Figure 4C D*). The release rate of the fast component ($1/\tau_1$) of the merged capacitance and deconvolution data showed no signs of saturation consistent with our previous analyses of each data set separately. Fitting a Hill equation to the merged data indicated a $K_D$ >50 µM and a Hill coefficient of 1.6 (*Figure 4C*). The release rate of the slow component ($1/\tau_2$; if existing) was on average more than 10 times smaller (black symbols, *Figure 4C*). These data indicate that there are at least two distinct kinetic steps contributing to release within the first 10 ms.

## Fast sustained release with very weak $Ca^{2+}$-dependence

To gain more insights into the mechanisms of sustained vesicle release, we focused on the synaptic release within the first 100 ms after $Ca^{2+}$ uncaging, presumably reflecting vesicle replenishment (*Sakaba, 2008*). Using capacitance measurements, we investigated the $Ca^{2+}$-dependence of sustained release by estimating the number of vesicles ($N_v$) released between 10 and 100 ms after flash onset (*Figure 5A*), assuming a single vesicle capacitance of 70 aF (*Hallermann et al., 2003*). There

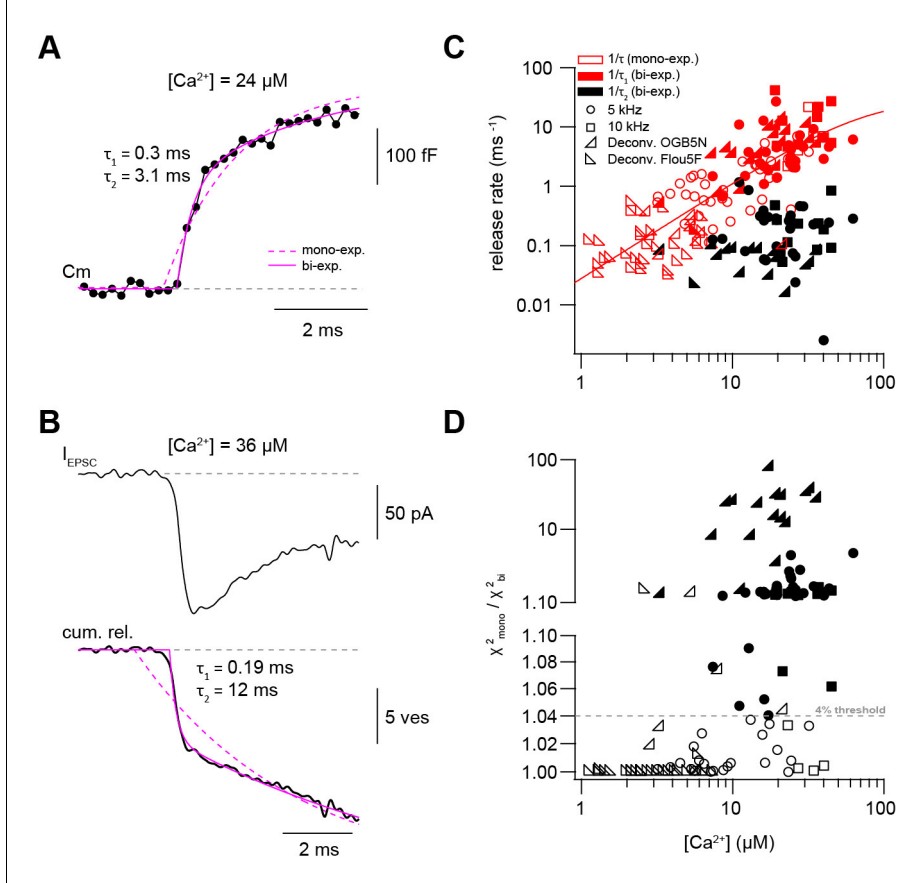

**Figure 4.** Presynaptic and postsynaptic measurements reveal two kinetic processes of neurotransmitter release. (A) Example of a capacitance trace showing the two components of release observed within the first 10 ms in response to UV-flash-evoked increase in $Ca^{2+}$ concentration to 24 μM. The solid magenta line represents the bi-exponential fit and the dashed magenta line represents mono-exponential fit (see *Equation 1*). (B) *Top:* example trace of an EPSC recording in response to UV-flash evoked increase in $Ca^{2+}$ concentration to 36 μM. *Bottom:* the corresponding cumulative release trace obtained from deconvolution analysis, showing the two components of release observed within the first 10 ms. The solid magenta line represents the bi-exponential fit and the dashed magenta line represents mono-exponential fit (see *Equation 1*). (C) Top: plot of neurotransmitter release rates as a function of peak $Ca^{2+}$ concentration (n = 80 and 59 capacitance measurements and deconvolution analysis, respectively). Data obtained from capacitance measurements with sinusoidal frequency of 5 kHz are shown as circles, data from 10 kHz capacitance measurements are shown as squares, and cumulative release data obtained from deconvolution analysis are shown as lower left- and lower right- triangles for recordings with OGB-5N and Fluo5F, respectively. Open symbols correspond to data from the mono-exponential fits and filled symbols correspond to data from the bi-exponential fits. Red symbols represent merged data of the release rates obtained from mono-exponential fit and the fast component of the bi-exponential fit, and black symbols represent the second component of the bi-exponential fit. The line represents a fit with a Hill equation with best-fit parameters $V_{max}$ = 29.9 $ms^{-1}$, $K_D$ = 75.5 μM, and $n$ = 1.61. (D) $\chi^2$ ratio for the mono-exponential compared to the bi-exponential fits. Dashed line represents the threshold of the $\chi^2$ ratio used to judge the fit quality of double compared to mono-exponential fits (as one criterion for selection). 5 kHz capacitance data are shown as circles, 10 kHz capacitance data are shown as squares, and cumulative release data (obtained from deconvolution analysis) are shown as lower left- and lower right- triangles for recordings with OGB-5N and Fluo5F, respectively. Open symbols correspond to data points judged as mono-exponential and filled symbols correspond to data points judged as bi-exponential.

The online version of this article includes the following source data for figure 4:

**Source data 1.** Presynaptic and postsynaptic measurements reveal two kinetic processes of neurotransmitter release.

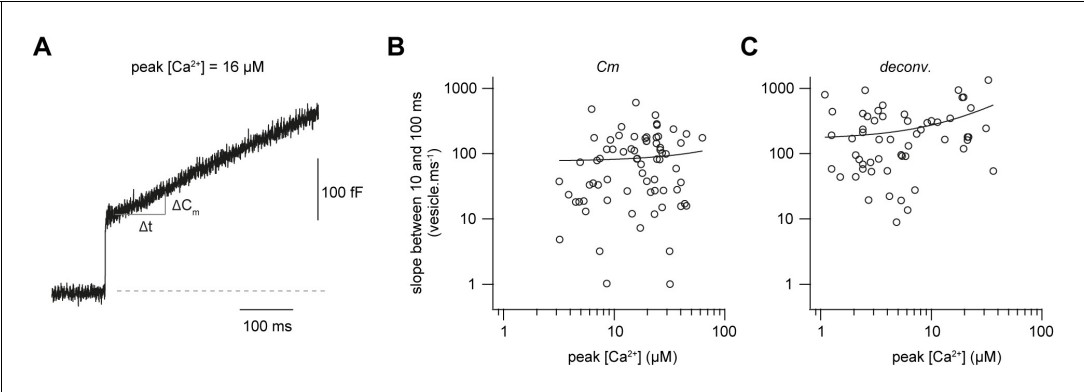

**Figure 5.** Fast sustained release with very weak Ca²⁺-dependence. (A) Examples of a capacitance trace showing a sustained component of release. (B) Plot of the number of vesicles released between 10 and 100 ms as estimated from capacitance measurements divided by the time interval (90 ms) versus the post-flash Ca²⁺ concentration (n = 71 cMFBs). The line represents a linear fit to the data with a slope of 6 vesicles $ms^{-1}$ $\mu M^{-1}$ (Pearson correlation coefficient = 0.06, $r^2$ = 0.003; $P_{Pearson\ correlation}$ = 0.6). (C) Plot of the number of vesicles released between 10 and 100 ms as estimated from deconvolution analysis divided by the time interval (90 ms) versus the post-flash Ca²⁺ concentration (n = 51 cMFB-GC pairs). The line represents a linear fit to the data with a slope of 10 vesicles $ms^{-1}$ $\mu M^{-1}$ (Pearson correlation coefficient = 0.3, $r^2$ = 0.1; $P_{Pearson\ correlation}$ = 0.01).
The online version of this article includes the following source data for figure 5:

**Source data 1.** Fast sustained release with very weak Ca²⁺-dependence.

was considerable variability in the release rate between 10 and 100 ms, which could be due to differences in bouton size and wash-out of proteins during whole-cell recordings. However, the release rate showed no obvious dependence on the post-flash Ca²⁺ concentration (*Figure 5B*). A comparable dose-response curve was obtained when investigating the rate of release between 10 and 100 ms using deconvolution analysis of postsynaptic currents, however, with a weak but significant correlation (*Figure 5C*). These data indicate that the slope of the sustained component of release is, if anything, weakly dependent on the intracellular Ca²⁺ concentration in the range of 1–50 µM, consistent with previously observed Ca²⁺-independent vesicle replenishment as assessed by depolarizing cMFBs to 0 mV in the absence or presence of intracellular EGTA (*Ritzau-Jost et al., 2014*).

## Release schemes with five Ca²⁺ steps and fast replenishment via parallel or sequential models can explain Ca²⁺-dependence of release

To investigate the mechanisms that could explain a non-saturating and shallow dose-response curve and rapid sustained release, we performed modeling with various release schemes. First, we simulated the exact time course of the concentration of free Ca²⁺. The Ca²⁺ release from DMn and subsequent binding to other buffers and the Ca²⁺ indicator were simulated based on previously described binding and unbinding rates (*Faas et al., 2005*; *Faas et al., 2007*; *Figure 6A*; *Table 2*; see Materials and methods). In contrast to previous results, which predicted a significant overshoot of Ca²⁺ following UV illumination with short laser pulses (*Bollmann et al., 2000*), our simulations predict little overshoot compared to the Ca²⁺ concentration measured by the Ca²⁺ indicator (*Figure 6B*). The discrepancy is readily described by recent improvements in the quantification of Ca²⁺ binding and unbinding kinetics (*Faas et al., 2005*; *Faas et al., 2007*). The calculations predict an almost step-like increase in the free Ca²⁺ concentration with a 10–90% rise time below 50 µs. These simulated UV illumination-induced transients of free Ca²⁺ concentrations were subsequently used to drive the release schemes. Realistic noise was added to the resulting simulated cumulative release rate and the traces were fit with exponential functions (*Equation 1*) as the experimental data (*Figure 6C*).

We compared three different release schemes in their ability to reproduce our experimental data. In model 1, a single pool of vesicles with two Ca²⁺ binding steps was used as previously established, for example for chromaffin cells and rod photoreceptors (*Duncan et al., 2010*; *Voets, 2000*). Such an assumption would readily explain the shallow dose-response curve (*Bornschein and Schmidt, 2018*). The 2nd component of release could be replicated by assuming rapid vesicle replenishment from a reserve pool ($V_R$; *Figure 6D*). However, adjusting the free parameters did not allow

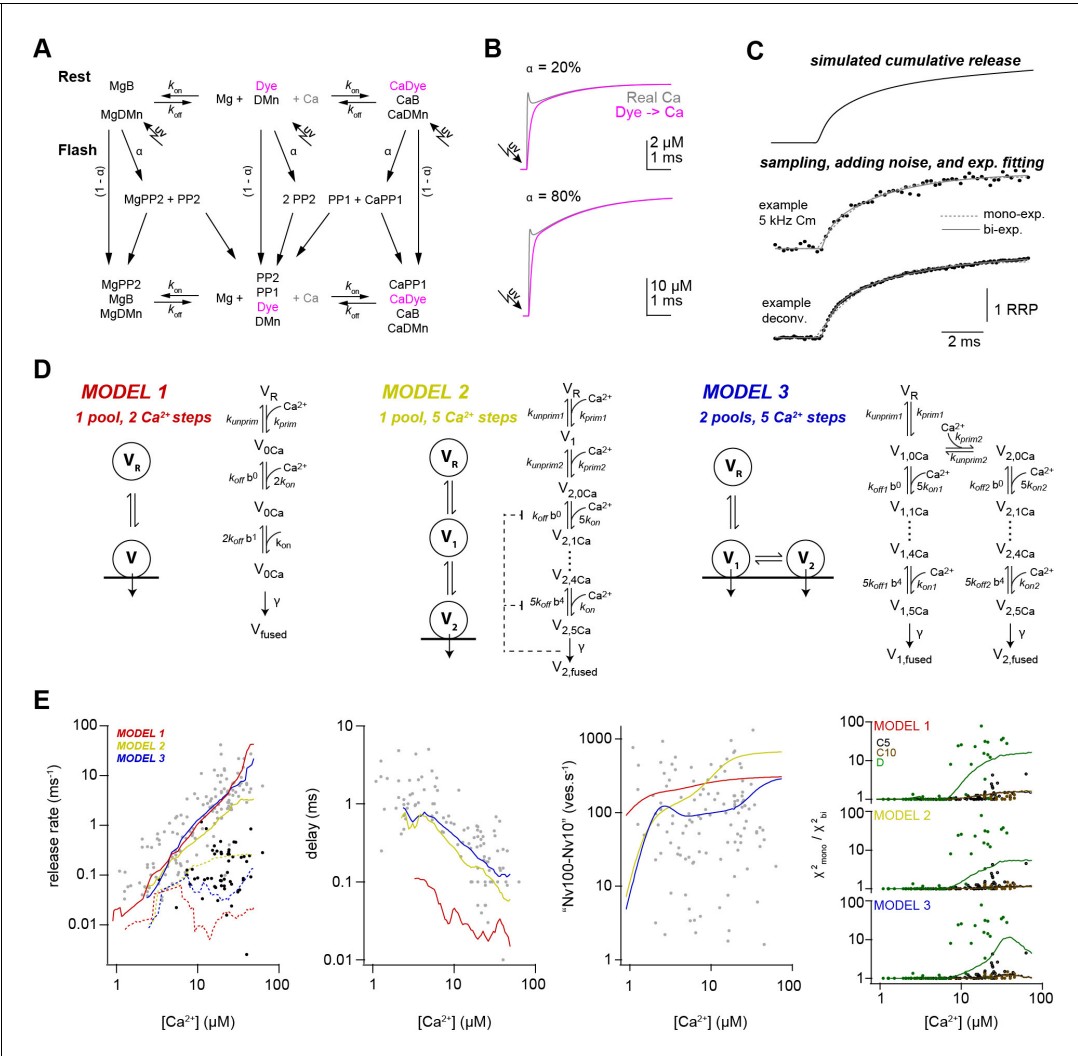

**Figure 6.** Release schemes with five $Ca^{2+}$ steps and fast replenishment via parallel or sequential models can explain $Ca^{2+}$-dependence of release. (**A**) Scheme of the chemical reactions that were implemented in the model to caluclate the UV-illumination-evoked increase in the free $Ca^{2+}$ concentration. The model considered $Ca^{2+}$ (Ca) and $Mg^{2+}$ (Mg) binding to the indicator dye (OGB-5N or Fluo-5F), to DM-nitrophen (DMn), and to buffers (ATP and/or an endogenous buffer). The forward ($k_{on}$) and backward ($k_{off}$) rate constants differ between chemical species. Upon photolysis, a fraction $\alpha$ of metal bound and free DMn made a transition to different photoproducts (PP1 and PP2; **Faas et al., 2005**). For model parameters see **Table 2**. (**B**) The scheme in (**A**) was converted to a system of differential equations and the time courses of the 'real' free $Ca^{2+}$ (magenta) and the free $Ca^{2+}$ reported by $Ca^{2+}$ dye were simulated for the indicated uncaging fractions $\alpha$. Note that after less than 1 ms the dye reliably reflects the time course of $Ca^{2+}$. (**C**) Traces showing the steps used in the simulation of the kinetic model of release. (**D**) Graphical illustration of the three models used during the simulations. For model parameters see **Table 3**. (**E**) From left to right, predictions of each model and the experimental data for the inverse of $\tau_1$ (gray symbols, solid lines) and inverse of $\tau_2$ (black symbols, dashed lines), delay, vesicle replenishment rate between 10 and 100 ms, and the increase in the $\chi^2$ ratio for the single- compared to the bi-exponential fits. Red, yellow, and blue lines correspond to simulations of models 1, 2, and 3, respectively. For the $\chi^2$ ratio (*right plot*), the experimental data and the simulations are shown separately for 5-kHz- and 10-kHz-capacitance data (C5 and C10; black and brown, respectively) and the deconvolution data (D; green).

reproducing the synaptic delay (**Figure 6E**). We therefore tested two more sophisticated models in which vesicle fusion is triggered via five $Ca^{2+}$ binding steps (**Schneggenburger and Neher, 2000**). In model 2, onevesicle pool represents the docked vesicles ($V_2$) and the other pool represents a replacement pool ($V_1$), which can undergo rapid docking and fusion (**Miki et al., 2016**; **Miki et al., 2018**), therefore representing two kinetic steps occurring in sequence. In model 3, two pools of vesicles ($V_1$, $V_2$) with different $Ca^{2+}$-sensitivity exist, where both types of vesicles can fuse with different $Ca^{2+}$ affinity (**Voets, 2000**; **Walter et al., 2013**; **Wölfel et al., 2007**; **Hallermann et al., 2010**) therefore representing two kinetic steps occurring in parallel. Model 3 reproduced the data

**Table 2.** Parameters for simulations of $Ca^{2+}$ release from DMN cage.

| Parameters | | Values | References number / Notes |
|---|---|---|---|
| Resting $Ca^{2+}$ | $[Ca^{2+}]_{rest}$ | $227*10^{-9}$ M | Measured |
| Total magnesium | $[Mg^{2+}]_T$ | $0.5*10^{-3}$ M | Pipette concentration |
| Fluo-5F | [Fluo] | 0 or 50 $*10^{-6}$ M (see **Table 1**) | Pipette concentration |
| | $K_D$ | $0.83*10^{-6}$ M | **Delvendahl et al., 2015** |
| | $k_{off}$ | 249 s-1 | ibid |
| | $k_{on}$ | $3*10^8$ $M^{-1}s^{-1}$ | **Yasuda et al., 2004** |
| OGB-5N | [OGB] | 0 or $200*10^{-6}$ M (see **Table 1**) | Pipette concentration |
| | $K_D$ | $31.4*10^{-6}$ M | Measured (**Figure 3—figure supplement 1A**) |
| | $k_{off}$ | 6000 $s^{-1}$ | ibid. |
| | $k_{on}$ | $2.5*10^8$ $M^{-1}s^{-1}$ | **DiGregorio and Vergara, 1997** |
| ATP | [ATP] | $5*10^{-3}$ M | Pipette concentration |
| $Ca^{2+}$ binding | $K_D$ | $2*10^{-4}$ M | **Meinrenken et al., 2002** |
| | $k_{off}$ | 100 000 $s^{-1}$ | ibid. |
| | $k_{on}$ | $5*10^8$ $M^{-1}s^{-1}$ | ibid. |
| $Mg^{2+}$ binding | $K_D$ | $100*10^{-6}$ M | **Bollmann et al., 2000**; MaxC |
| | $k_{off}$ | 1000 $s^{-1}$ | ibid. |
| | $k_{on}$ | $1*10^7$ $M^{-1}s^{-1}$ | ibid. |
| Endogenous buffer | [EB] | $480*10^{-6}$ M | **Delvendahl et al., 2015** |
| | $K_D$ | $32*10^{-6}$ M | ibid |
| | $k_{off}$ | 16 000 $s^{-1}$ | ibid. |
| | $k_{on}$ | $5*10^8$ $M^{-1}s^{-1}$ | ibid. |
| Total DM nitrophen | $[DMn]_T$ | $500*10^{-6}$ – $10*10^{-3}$ M (see **Table 1**) | Pipette concentration |
| $Ca^{2+}$ binding | $K_D$ | $6.5*10^{-9}$ M | **Faas et al., 2005** |
| | $k_{off}$ | 0.19 $s^{-1}$ | ibid. |
| | $k_{on}$ | $2.9*10^7$ $M^{-1}s^{-1}$ | ibid. |
| $Mg^{2+}$ binding | $K_D$ | $1.5*10^{-6}$ M | ibid. |
| | $k_{off}$ | 0.2 $s^{-1}$ | ibid. |
| Uncaging fraction | α | See **Table 1** | |
| Fast uncaging fraction | af | 0.67 | **Faas et al., 2005** |
| Photoproduct 1 | [PP1] | | |
| $Ca^{2+}$ binding | $K_D$ | $2.38*10^{-3}$ M | **Faas et al., 2005** |
| | $k_{off}$ | 69 000 $s^{-1}$ | ibid. |
| | $k_{on}$ | $2.9*10^7$ $M^{-1}s^{-1}$ | ibid. |
| $Mg^{2+}$ binding | $K_D$ | $1.5*10^{-6}$ M | ibid. |
| | $k_{off}$ | 300 $s^{-1}$ | ibid. |
| | $k_{on}$ | $1.3*10^5$ $M^{-1}s^{-1}$ | ibid. |
| Photoproduct 2 | [PP2] | | |
| $Ca^{2+}$ binding | $K_D$ | $124.1*10^{-6}$ M | Ibid. |
| | $k_{off}$ | 3600 $s^{-1}$ | ibid. |
| | $k_{on}$ | $2.9*10^7$ $M^{-1}s^{-1}$ | ibid. |
| $Mg^{2+}$ binding | $K_D$ | $1.5*10^{-6}$ M | ibid. |
| | $k_{off}$ | 300 $s^{-1}$ | ibid. |
| | $k_{on}$ | $1.3*10^5$ $M^{-1}s^{-1}$ | ibid. |

**Table 3.** Parameters for release scheme models.

| Model1 | | Model2 | | Model3 | |
|---|---|---|---|---|---|
| $k_{on}$ | $2.95*10^9$ Ca$^{2+}$(t) M$^{-1}$ s$^{-1}$ | $k_{on,init}$ | $5.10*10^8$ Ca$^{2+}$(t) M$^{-1}$ s$^{-1}$ | $k_{on1}$ | $0.5\ k_{on2}$ |
| | | $k_{on,plug}$ | $0.1\ k_{on,init}$ | $k_{on2}$ | $5.10*10^8$ Ca$^{2+}$(t) M$^{-1}$ s$^{-1}$ |
| $k_{off}$ | $4.42*10^5$ s$^{-1}$ | $k_{off,init}$ | $2.55*10^4$ s$^{-1}$ | $k_{off1}$ | $10\ k_{off2}$ |
| | | $k_{off,plug}$ | $0.4\ k_{off,init}$ | $k_{off2}$ | $2.55*10^4$ s$^{-1}$ |
| b | 0.25 | b | 0.25 | b | 0.25 |
| $\gamma$ | $1.77*10^4$ s$^{-1}$ | $\gamma$ | $1.77*10^4$ s$^{-1}$ | $\gamma$ | $1.77*10^4$ s$^{-1}$ |
| $k_{prim}$ | $0.6+30*(Ca^{2+}(t)/(K_{D,prim}+Ca^{2+}(t)))$ s$^{-1}$ | $k_{prim1}$ | $2.5+60*(Ca^{2+}(t)/(K_{D,prim1}+Ca^{2+}(t)))$ s$^{-1}$ | $k_{prim1}$ | 30 s$^{-1}$ |
| $k_{unprim}$ | $0.6+30*(Ca^{2+}_{Rest}/(K_{D,prim}+Ca^{2+}_{Rest}))$ s$^{-1}$ | $k_{unprim1}$ | $2.5+60*(Ca^{2+}_{Rest}/(K_{D,prim1}+Ca^{2+}_{Rest}))$ s$^{-1}$ | $k_{unprim1}$ | 30 s$^{-1}$ |
| $K_{D,prim}$ | 2 µM | $K_{D,prim1}$ | 2 µM | | |
| | | $k_{prim2}$ | $100+800*(Ca^{2+}(t)/(K_{D,prim2}+Ca^{2+}(t)))$ s$^{-1}$ | $k_{prim2}$ | $0.5+30*(Ca^{2+}(t)/(K_{D,prim2}+Ca^{2+}(t)))$ s$^{-1}$ |
| | | $k_{unprim2}$ | $100+800*(Ca^{2+}_{Rest}/(K_{D,prim2}+Ca^{2+}_{Rest}))$ s$^{-1}$ | $k_{unprim2}$ | $0.5+30*(Ca^{2+}_{Rest}/(K_{D,prim2}+Ca^{2+}_{Rest}))$ s$^{-1}$ |
| | | $K_{D,prim2}$ | 2 µM | $K_{D,prim2}$ | 2 µM |

as good as model 2; however, the non-saturation up to 50 µM could be reproduced somewhat better in model 3. Interestingly, models 2 and 3 both replicated the observed shallow dose-response curve despite the presence of five Ca$^{2+}$ binding steps. These results indicate that established models with five Ca$^{2+}$-steps incorporating fast vesicle replenishment via sequential or parallel vesicle pools can replicate our data fairly well.

## Ca$^{2+}$ uncaging with different pre-flash Ca$^{2+}$ concentrations indicates Ca$^{2+}$-dependent vesicle priming

Finally, we aimed to obtain a mechanistic understanding that could explain both the strong dependence of action potential-evoked release on basal Ca$^{2+}$ concentration (*Figure 1*) and the Ca$^{2+}$-dependence of vesicle fusion (*Figures 2–6*). In principle, the action potential-evoked data in *Figure 1* could be explained by an acceleration of vesicle fusion kinetics or, alternatively, an increase in the number of release-ready vesicles upon elevated basal Ca$^{2+}$. To differentiate between these two mechanistic possibilities, we investigated the effect of basal Ca$^{2+}$ concentration preceding the UV illumination (pre-flash Ca$^{2+}$) on flash-evoked release. The pre-flash Ca$^{2+}$ concentration can only be reliably determined with the Ca$^{2+}$ indicator Fluo5F used in the experiments with weak flashes (see *Table 1*). We therefore grouped the deconvolution experiments with weak flashes, which elevated the Ca$^{2+}$ concentration to less than 7 µM, into two equally sized groups of low and high pre-flash Ca$^{2+}$ (below and above a value of 200 nM, respectively). Due to the presence of the Ca$^{2+}$ loaded DMn cage, the pre-flash Ca$^{2+}$ concentrations were on average higher than the resting Ca$^{2+}$ concentration in physiological conditions of around 50 nM (*Delvendahl et al., 2015*). In both groups, the post-flash Ca$^{2+}$ concentration was on average similar (~3 µM; *Figure 7B*). The peak EPSC amplitude of postsynaptic current was significantly larger with high compared to low pre-flash Ca$^{2+}$ concentration (38 ± 10 and 91 ± 16 pA, n = 18 and 13, respectively, P$_{Mann-Whitney}$ = 0.001; *Figure 7A and C*). Correspondingly, the amplitude of the fast component of release as measured from deconvolution analysis was larger with high compared to low pre-flash Ca$^{2+}$ (18 ± 5 and 49 ± 10, n = 18 and 13, respectively, P$_{Mann-Whitney}$ = 0.005; *Figure 7C*). However, the kinetics of vesicle fusion, measured as the inverse of the time constant of the fast component of release, were not significantly different for both conditions (0.15 ± 0.04 and 0.12 ± 0.03 ms$^{-1}$ for the low and high pre-flash Ca$^{2+}$ conditions, n = 18 and 13, respectively, P$_{Mann-Whitney}$ = 0.74; *Figure 7C*). The delay was also not significantly different (P$_{Mann-Whitney}$ = 0.54; *Figure 7C*). These data indicate that the number of release-ready vesicles were increased upon elevating the basal Ca$^{2+}$ concentration but the fusion kinetics were unaltered. We therefore added an additional Ca$^{2+}$-dependent maturation step to the initial vesicle priming of the release schemes (see Materials and methods; note that this was already present in the above-described simulations of *Figure 6* but it has little impact on these data). This allowed replicating the

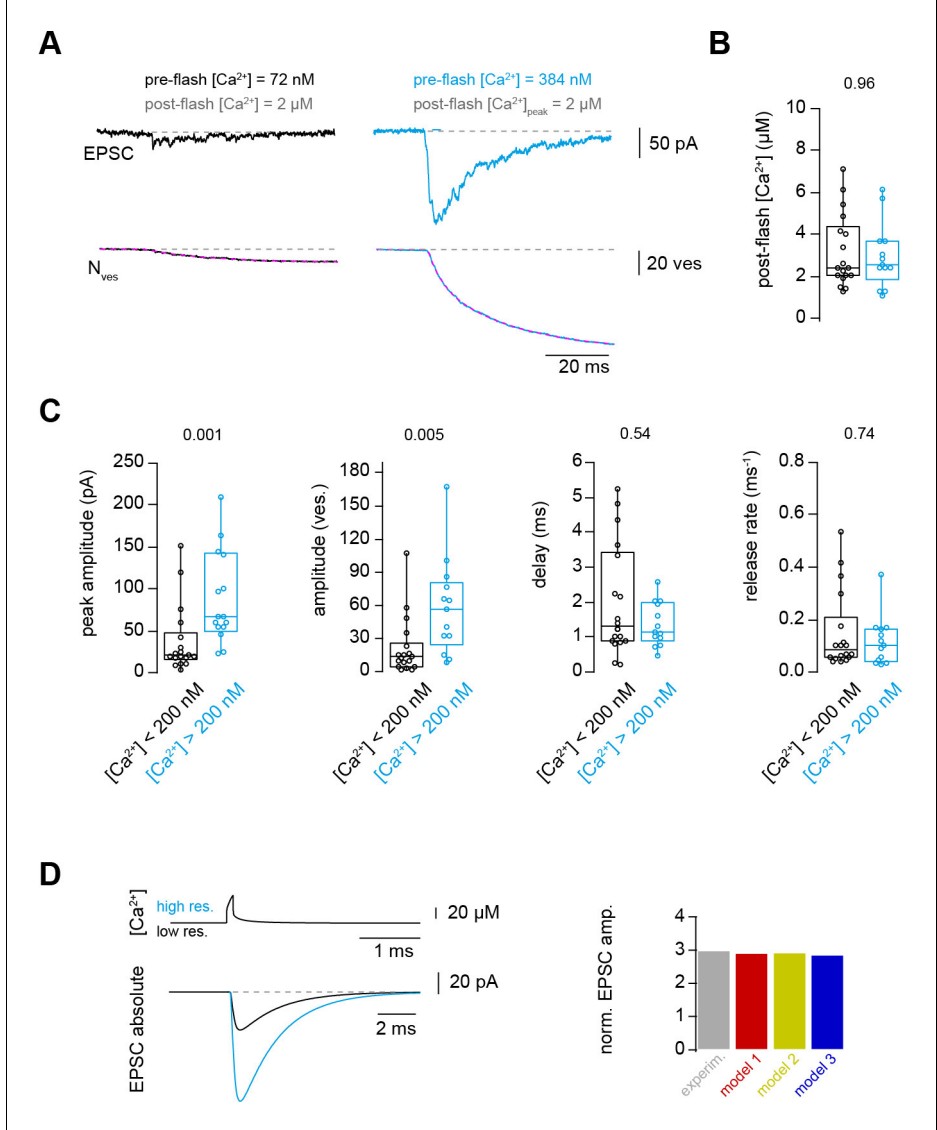

**Figure 7.** Ca²⁺ uncaging with different pre-flash Ca²⁺ concentrations indicates Ca²⁺-dependent vesicle priming.
(**A**) Two consecutive recordings from the same cell pair, with the same post-flash Ca²⁺ concentration but different pre-flash Ca²⁺ concentration in the presynaptic terminal. *Top:* postsynaptic current. *Bottom:* cumulative release of synaptic vesicles measured by deconvolution analysis of EPSCs superposed with a mono-exponential fit (magenta). Black and blue color represent low and high pre-flash Ca²⁺ concentration, respectively. The pre- and post-flash Ca²⁺ concentrations are indicated in each panel. (**B**) Comparison of the average post-flash Ca²⁺ concentration between both groups of either low (black) or high (blue) pre-flash Ca²⁺ concentration (n = 18 and 13 pairs, respectively). (**C**) From left to right: comparisons of the peak amplitude, the number of released vesicles measured as obtained from deconvolution analysis of EPSC, the delay of the release onset, and the release rate. Boxplots show median and 1st/3rd quartiles with whiskers indicating the whole data range. The values above the boxplots represent P-values of Mann-Whitney *U* tests. (**D**) *Top left:* simulated local action potential-evoked Ca²⁺ concentrations at 20 nm from a Ca²⁺ channel taken from ***Delvendahl et al., 2015***. Note the almost complete overlap of the two Ca²⁺ concertation traces with low and high basal Ca²⁺ concertation. *Bottom left:* predicted action potential-evoked EPSCs with low and high basal Ca²⁺ concertations. *Right:* ratio of the action potential-evoked EPSC amplitude with high and low basal Ca²⁺ concentrations for the experimental data and the model predictions.

The online version of this article includes the following source data for figure 7:

**Source data 1.** Ca²⁺ uncaging with different pre-flash Ca²⁺ concentrations indicates Ca²⁺-dependent vesicle priming.

three-fold increase in the action potential-evoked release when driving the release scheme with a previously estimated local $Ca^{2+}$ concentration during an action potential (*Figure 7D*; *Delvendahl et al., 2015*). Thus, the release schemes 2 and 3 describe all our experimental data and therefore represent to our knowledge the only release scheme explaining the priming, fusion, and replenishment of vesicles at a mature excitatory synapse in the CNS at physiological temperature.

## Discussion

Here, we provided insights into the $Ca^{2+}$-dependence of vesicle priming, fusion, and replenishment at cMFBs. The results obtained at this synapse show prominent $Ca^{2+}$-dependent priming steps, a shallow non-saturating dose-response curve up to 50 μM, and little $Ca^{2+}$-dependence of sustained vesicle replenishment. Our computational analysis indicates that the peculiar dose-response curve can be explained by well-established release schemes having five $Ca^{2+}$ steps and rapid vesicle replenishment via sequential or parallel vesicle pools. Thus, we established quantitative scheme of synaptic release for a mature high-fidelity synapse, exhibiting both high- and low-affinity $Ca^{2+}$ sensors.

### $Ca^{2+}$ affinity of the vesicle fusion sensor

The $Ca^{2+}$-sensitivity of vesicle fusion seems to be synapse-specific. In contrast to the estimated $Ca^{2+}$ affinity for vesicle fusion of ~100 μM at the bipolar cell of goldfish (*Heidelberger et al., 1994*) and the squid giant synapse (*Adler et al., 1991*; *Llinás et al., 1992*), recent studies showed that the affinity is much higher at three types of mammalian central synapses: the calyx of Held (*Bollmann et al., 2000*; *Lou et al., 2005*; *Schneggenburger and Neher, 2000*; *Sun et al., 2007*; *Wang et al., 2008*), the inhibitory cerebellar basket cell to Purkinje cell synapse (*Sakaba, 2008*), and the hippocampal mossy fiber boutons (*Fukaya et al., 2021*). Consistent with reports from mammalian central synapses, our data revealed prominent vesicle fusion at concentrations below 5 μM arguing for a high-affinity fusion sensor (*Figures 2–4*). However, the non-saturation of the dose-response curve (*Figures 2–4*) argues for the presence of a rather low-affinity fusion sensor at cMFBs. In our simulations, both models 2 and 3 exhibit vesicles with a $Ca^{2+}$-affinity similar to the calyx of Held. Nevertheless, with high intracellular $Ca^{2+}$ concentrations (>20 μM) these vesicles will fuse very rapidly and the further increase in the release kinetics (causing the non-saturating dose-response curve) can be explained by rapid vesicle replenishment from a sequential pool of vesicles exhibiting use-dependent lowering of the $Ca^{2+}$-affinity ($V_1$ in model 2; *Miki et al., 2018*) or from a parallel pool of vesicles with lower $Ca^{2+}$ affinity ($V_1$ in model 3; *Hallermann et al., 2010*). Our data therefore indicate that the shallow and non-saturating dose-response curve is the consequence of rapid replenishment of vesicles that still exhibit a lower $Ca^{2+}$-affinity compared to fully recovered vesicles. Consistent with this interpretation, a lowering in the $Ca^{2+}$-affinity of the vesicle fusion sensor has been observed at the calyx of Held with $Ca^{2+}$ uncaging following vesicle depletion (*Müller et al., 2010*; *Wadel et al., 2007*). These newly replenished vesicles might contribute particularly to the dose-response curve at the cMFB because the cMFB has a much faster rate of vesicle replenishment compared with the calyx of Held synapse (*Miki et al., 2020*) providing a possible explanation why the here-reported dose-response curve differs from previous results at the calyx of Held. Furthermore, cMFBs seem to have functional similarities with ribbon-type synapses because it has recently been shown that the vesicle mobility in cMFBs is comparable to ribbon-type synapses (*Rothman et al., 2016*). The hallmark of ribbon-type synapses is their rapid vesicle replenishment (*Lenzi and von Gersdorff, 2001*; *Matthews, 2000*) and indeed more shallow dose-response curves were obtained at the ribbon photoreceptors and inner hair cell synapses (*Duncan et al., 2010*; *Heil and Neubauer, 2010*; *Johnson et al., 2010*; *Thoreson et al., 2004*, but see *Beutner et al., 2001*). The newly replenished vesicles might be molecularly immature and resemble vesicles that have only a near-linear remaining $Ca^{2+}$ sensor when the fast $Ca^{2+}$ sensor synaptotagmin II is lacking (*Kochubey and Schneggenburger, 2011*).

The here obtained dose-response curve has the following three caveats. First, the cMFB to GC synapses in lobule IX are functionally distinct based on the origin of the mossy fibers (*Chabrol et al., 2015*). Therefore, the here-recorded boutons in lobule IV/V could be molecularly and functionally distinct leading to the observed scatter in the dose-response curve, which could cause an apparent shallowing. Yet, the degree of scatter in the vesicular release rate at the cMFB seems comparable to

studies at other synapses (*Fukaya et al., 2021*; *Heidelberger et al., 1994*; *Sakaba, 2008*) including the calyx of Held (*Bollmann et al., 2000*; *Schneggenburger and Neher, 2000*), although the functional heterogeneity between different types of calyces (*Grande and Wang, 2011*) could be explained by differences in the coupling distance (*Fekete et al., 2019*). Second, we could not investigate allosteric or two-sensor models (*Lou et al., 2005*; *Sun et al., 2007*; *Li et al., 2021*) because we did not address the release rates in the low $Ca^{2+}$ range ($<1$ µM), therefore, these questions remain to be investigated at the cMFBs. Third, currently available techniques to estimate fast release rates at near-physiological temperatures in the $Ca^{2+}$ range above 50 µM are limited by the sampling frequency of capacitance measurements and dendritic filtering, which could prevent the detection of saturation at the upper end of the dose-response curve.

## $Ca^{2+}$-sensitivity of vesicle priming

In previous reports, the $Ca^{2+}$-dependence of vesicle priming and replenishment at cMFBs was analyzed more indirectly with the $Ca^{2+}$ chelator EGTA (*Ritzau-Jost et al., 2014*; *Ritzau-Jost et al., 2018*) and the obtained results could be explained by $Ca^{2+}$-dependent models but surprisingly also by $Ca^{2+}$-independent models (*Hallermann et al., 2010*; *Ritzau-Jost et al., 2018*). Furthermore, the analysis of molecular pathways showed that the recovery from depression is independent of the $Ca^{2+}$/calmodulin/Munc13 pathway at cMFBs (*Ritzau-Jost et al., 2018*). Our paired recordings and uncaging experiments (*Figures 1* and *7*) clearly demonstrate pronounced $Ca^{2+}$-dependence of vesicle priming at cMFBs. Taken together, these data indicate that some priming steps are mediated by $Ca^{2+}$-dependent mechanisms, which do not involve the $Ca^{2+}$/calmodulin/Munc13 pathway. A potential candidate for such a $Ca^{2+}$-dependent mechanism are the interaction of diacylgylcerol/phospholipase C or $Ca^{2+}$/phospholipids with Munc13s (*Lee et al., 2013*; *Lou et al., 2008*; *Rhee et al., 2002*; *Shin et al., 2010*). Another candidate for a high-affinity $Ca^{2+}$ sensor is Synaptotamin 7 (see below).

Synaptic vesicles that fuse upon single action potentials (*Figure 1*) and weak uncaging stimuli (post-flash $Ca^{2+}$ concentration of ~3 µM; *Figure 7*) are particularly fusogenic and thus might represent the superprimed vesicles with a particular high release probability (*Hanse and Gustafsson, 2001*; *Ishiyama et al., 2014*; *Kusch et al., 2018*; *Lee et al., 2013*; *Schlüter et al., 2006*; *Taschenberger et al., 2016*) suggesting that the process of superpriming is $Ca^{2+}$-dependent. This interpretation would also provide an explanation why in a recent report, triggering an action potential in the range of 10–50 ms before another action potential restored the synchronicity of synaptic vesicle fusion in mutant synapses which had an impaired synchronous release (*Chang et al., 2018*). It would be furthermore consistent with a proposed rapid, dynamic, and $Ca^{2+}$-dependent equilibrium between primed and superprimed vesicles (*Neher and Brose, 2018*). However, further investigations are needed for the dissection between the $Ca^{2+}$-dependence of priming and superpriming. Yet, our data show that some priming steps are strongly $Ca^{2+}$-dependent with a high-affinity $Ca^{2+}$ sensor that allow detecting changes between 30 and 180 nM at cMFBs.

## $Ca^{2+}$-sensitivity of vesicle replenishment

The upstream steps of vesicle priming, referred to as replenishment, recruitment, refilling, or reloading, remain controversial in particular with respect to their speed. The slow component of release (during prolonged depolarizations or $Ca^{2+}$ elevations with uncaging) was initially interpreted as a sub-pool of release-ready vesicles that fuse with slower kinetics (see e.g. *Sakaba and Neher, 2001a*). However, recent studies indicate very fast vesicle replenishment steps (*Blanchard et al., 2020*; *Chang et al., 2018*; *Doussau et al., 2017*; *Hallermann et al., 2010*; *Lee et al., 2012*; *Malagon et al., 2020*; *Miki et al., 2016*; *Miki et al., 2018*; *Saviane and Silver, 2006*; *Valera et al., 2012*). These findings further complicate the dissection between fusion, priming, and replenishment steps. Therefore, the differentiation between 'parallel' release schemes with fast and slowly fusing vesicles and 'sequential' release schemes with fast vesicle replenishment and subsequent fusion is technically challenging at central synapses. Our data could be described by both sequential and parallel release schemes (models 2 and 3; *Figure 6*). The non-saturation of the release rate could be described somewhat better by the parallel model 3. However, further adjustment of the use-dependent slowing of the rates in model 2 (see $k_{on,plug}$, $k_{off,plug}$, *Equations 3 and 4*; *Miki et al., 2018*) can result in a sequential model exhibiting both fast and slowly fusing vesicles with different $Ca^{2+}$-sensitivity (see *Mahfooz et al., 2016*, for an alternative description of use-dependence of vesicle fusion).

Such use-dependent sequential models ultimately complicate the semantic definitions of 'sequential' and 'parallel', because the replenished vesicles of such sequential models will fuse in a molecularly different state, which could also be viewed as a parallel pathway to reach fusion. Independent of the difficulty to differentiate between sequential and parallel release schemes, the sustained component of release exhibited little $Ca^{2+}$-dependence in the here-tested range between 1 and 50 μM (*Figure 5*). However, it should be mentioned that measuring the sustained release rate is prone to errors with both presynaptic capacitance and postsynaptic current recordings, because the former cannot differentiate between exo- and endocytosis occurring simultaneously, and the latter can fail to dissect direct release from spill-over current, which is prominent at this synapse (*DiGregorio et al., 2002*). Nevertheless, the $Ca^{2+}$-independence of vesicle replenishment observed with capacitance measurements and the very weak $Ca^{2+}$-dependence observed with postsynaptic techniques seem consistent with the previously observed EGTA-independent slope of the sustained release during prolonged depolarizations (*Ritzau-Jost et al., 2014*). Our data cannot differentiate if replenishment is mediated by a saturated $Ca^{2+}$ sensor for priming (model 2; assumed $K_{dD}$ of 2 μM; *Miki et al., 2018*) or a parallel $Ca^{2+}$-independent step (model 3). Thus, during sustained activity at cMFBs, vesicle replenishment is mediated by either an apparently $Ca^{2+}$-independent process because of a saturated high-affinity $Ca^{2+}$ sensor or a $Ca^{2+}$-independent process.

## Implications for coupling distance

The $Ca^{2+}$-sensitivity of vesicle fusion critically impacts the estimates of the coupling distance between $Ca^{2+}$ channels and synaptic vesicles, mainly those obtained based on functional approaches (*Neher, 1998*; *Eggermann et al., 2011*; but not on structural approaches, see e.g. *Éltes et al., 2017*; *Rebola et al., 2019*). Our previous estimate of the coupling distance at the cMFB of 20 nm (*Delvendahl et al., 2015*) was based on the release scheme of *Wang et al., 2008* obtained at the calyx of Held synapse at an age of (P16-P19) at room temperature and assuming a $Q_{10}$ factor of 2.5. The now estimated $k_{on}$ and $k_{off}$ rates at mature cMFBs at physiological temperature were slightly larger and smaller than the temperature-corrected values from the calyx, respectively, resulting in a slightly higher affinity of the fast releasing vesicles ($V_2$ in models 2 and 3). Therefore, at the cMFB, the coupling distance of the vesicles released by a single action potential is if anything even smaller than the previous estimate of 20 nm.

## Implications for synaptic facilitation

Our data might contribute to a better understanding of the mechanisms of the 'residual $Ca^{2+}$ hypothesis' explaining synaptic facilitation (*Jackman and Regehr, 2017*; *Katz and Miledi, 1968*; *Magleby, 1987*; *Zucker and Regehr, 2002*). The strong dependence of the action potential-evoked release on basal $Ca^{2+}$ (*Figure 1*) supports the critical effect of residual $Ca^{2+}$ on synaptic strength. Our mechanistic analysis (particularly *Figure 7*) indicates that the number of release-ready vesicles rather than the vesicular release probability is regulated by residual $Ca^{2+}$. The high-affinity $Ca^{2+}$ sensor Synaptotagmin-7 (*Sugita et al., 2002*) could be a sensor for the changes in basal $Ca^{2+}$ levels and mediate the here-reported three-fold increase in synaptic strength (*Figures 1* and *7*). Synaptotagmin-7 has been shown to mediate vesicle recruitment (*Liu et al., 2014*), asynchronous release (*Luo and Südhof, 2017*), and synaptic facilitation (*Chen et al., 2017*; *Jackman et al., 2016*). If the recruitment and priming steps are fast enough they could provide a powerful mechanism for synaptic facilitation. Indeed, there is increasing evidence for ultra-fast $Ca^{2+}$-dependent recruitment and priming (reviewed in *Neher and Brose, 2018*) as well as facilitation mediated by an increase in the number of release-ready vesicles rather than the vesicular release probability (*Jackman et al., 2016*; *Kobbersmed et al., 2020*; *Vevea et al., 2021*). Our data are therefore consistent with the emerging view that facilitation is mediated by rapid Synaptotagmin-7/$Ca^{2+}$-dependent recruitment and priming of vesicles.

## Implications for high-frequency transmission

Synaptic fidelity has been shown to increase with age at cMFBs (*Cathala et al., 2003*), neocortical synapses (*Bornschein et al., 2019*), and the calyx of Held (*Fedchyshyn and Wang, 2005*; *Nakamura et al., 2015*; *Taschenberger and von Gersdorff, 2000*). During high-frequency transmission, the residual $Ca^{2+}$ concentration increases up to a few μM at cMFBs (*Delvendahl et al., 2015*)

but mature cMFBs can still sustain synchronous release (*Hallermann et al., 2010*; *Saviane and Silver, 2006*). The developmental decrease in the affinity of the release sensors observed at the calyx of Held (*Wang et al., 2008*) and the here-reported shallow-dose-response curve at mature cMFBs could be an evolutionary adaptation of synapses to prevent the depletion of the release-ready vesicles at medium $Ca^{2+}$ concentrations and therefore allow maintaining sustained synchronous neurotransmission with high fidelity (*Matthews, 2000*).

# Materials and methods

## Key resources table

| Reagent type (species) or resource | Designation | Source or reference | Identifiers | Additional information |
|---|---|---|---|---|
| Chemical compound, drug | NaCl | Sigma-Aldrich | Cat. # S9888 | |
| Chemical compound, drug | $NaHCO_3$ | Sigma-Aldrich | Cat. # S6297 | |
| Chemical compound, drug | Glucose | Sigma-Aldrich | Cat. # G8270 | |
| Chemical compound, drug | AP 5 | Sigma-Aldrich | Cat. # A78403 | |
| Chemical compound, drug | KCl | Sigma-Aldrich | Cat. # P9333 | |
| Chemical compound, drug | $CaCl_2$ | Sigma-Aldrich | Cat. # C5080 | For extracellular solution |
| Chemical compound, drug | $CaCl_2$ | Sigma-Aldrich | Cat. # 21115 | For intracellular solution |
| Chemical compound, drug | EGTA | Sigma-Aldrich | Cat. # E0396 | |
| Chemical compound, drug | $NaH_2PO_4$ | Merck | Cat. # 106342 | |
| Chemical compound, drug | Tetrodotoxin | Tocris | Cat. # 1078 | |
| Chemical compound, drug | $MgCl_2$ | Sigma-Aldrich | Cat. # M2670 | |
| Chemical compound, drug | TEA-Cl | Sigma-Aldrich | Cat. # T2265 | |
| Chemical compound, drug | HEPES | Sigma-Aldrich | Cat. # H3375 | |
| Chemical compound, drug | NaGTP | Sigma-Aldrich | Cat. # G8877 | |
| Chemical compound, drug | $Na_2ATP$ | Sigma-Aldrich | Cat. # A2383 | |
| Chemical compound, drug | DMnitrophen | Synptic systems | Cat. # 510016 | |
| Chemical compound, drug | CsOH | Sigma-Aldrich | Cat. # C8518 | |
| Chemical compound, drug | Atto594 | ATTO-TEC | Cat. # AD 594 | |
| Chemical compound, drug | OGB1 | Thermo Fisher Scientific | Cat. # 06806 | |
| Chemical compound, drug | OGB-5N | Thermo Fisher Scientific | Cat. # 944034 | |
| Chemical compound, drug | Fluo-5F | Thermo Fisher Scientific | Cat. # F14221 | |

*Continued on next page*

*Continued*

| Reagent type (species) or resource | Designation | Source or reference | Identifiers | Additional information |
|---|---|---|---|---|
| Chemical compound, drug | KOH solution | Roth | Cat. # K017.1 | |
| Chemical compound, drug | Kynurenic acid | Sigma-Aldrich | Cat. # K3375 | |
| Chemical compound, drug | Cyclothiazide | Sigma-Aldrich | Cat. # C9847 | |
| Chemical compound, drug | $Ca^{2+}$ Calibration Buffer Kit | Thermo Fisher Scientific | Cat. # C3008MP | |
| Chemical compound, drug | Caged fluorescein | Sigma-Aldrich | Cat. # F7103 | |
| Chemical compound, drug | Glycerol | Sigma-Aldrich | Cat. # G5516 | |
| Chemical compound, drug | Isoflourane | Baxter | Cat. # Hdg9623 | |
| Chemical compound, drug | Aqua B. Braun | Braun | Cat. # 00882479E | For extracellular solution |
| Chemical compound, drug | Sterile Water | Sigma-Aldrich | W Cat. # 3500 | For intracellular solution |
| Strain, strain background (mouse C57BL/6N) | Female, male C57BL/6N | Charles river | https://www.criver.com/ | |
| Other | Vibratome | LEICA VT 1200 | https://www.leica-microsystems.com/ | |
| Other | Femto2D laser-scanning microscope | Femtonics | https://femtonics.eu/ | |
| Other | UV laser source | Rapp OptoElectronic | https://rapp-opto.com/ | 375 nm, 200 mW |
| Other | DMZ Zeitz Puller | Zeitz | https://www.zeitz-puller.com/ | |
| Other | Borocilicate glass | Science Products | https://science-products.com/en/ | GB200F-10 With filament |
| Other | HEKA EPC10/2 amplifier | HEKA Elektronik | https://www.heka.com/ | |
| Other | Ti:Sapphire laser | MaiTai, SpectraPhysics | https://www.spectra-physics.com/ | |
| Other | $Ca^{2+}$ sensitive electrode (ELIT 8041 PVC membrane) | NICO 2000 | http://www.nico2000.net/index.htm | |
| Other | Single junction silver chloride reference electrode (ELIT 001 n) | NICO 2000 | http://www.nico2000.net/index.htm | |
| Other | PH/Voltmeter | Metler toledo | https://www.mt.com/de/en/home.html | |
| Other | Osmomat 3000 | Gonotec | http://www.gonotec.com/de | |
| Other | TC-324B perfusion heat controller | Warner Instruments | https://www.warneronline.com/ | |
| Software, algorithm | MES | Femtonics | https://femtonics.eu/ | |
| Software, algorithm | Igor Pro | Wavemetrics | https://www.wavemetrics.com/ | |
| Software, algorithm | Patchmaster | HEKA Elektronik | https://www.heka.com/ | |
| Software, algorithm | Adobe illustrator | Adobe | https://www.adobe.com/products/illustrator.html | |
| Software, algorithm | Mathematica | Wolfram | https://www.wolfram.com/mathematica/ | |
| Software, algorithm | Maxchelator | Stanford University | https://somapp.ucdmc.ucdavis.edu/pharmacology/bers/maxchelator/ | |

## Preparation

Animals were treated in accordance with the German Protection of Animals Act and with the guidelines for the welfare of experimental animals issued by the European Communities Council Directive. Acute cerebellar slices were prepared from mature P35–P42 C57BL/6 mice of either sex as previously described (*Hallermann et al., 2010*). Isoflurane was used to anesthetize the mice, which were then sacrificed by decapitation. The cerebellar vermis was quickly removed and mounted in a chamber filled with chilled extracellular solution. 300-µm-thick parasagittal slices were cut using a Leica VT1200 microtome (Leica Microsystems), transferred to an incubation chamber at 35°C for ~30 min, and then stored at room temperature until use. The extracellular solution for slice cutting and storage contained (in mM) the following: NaCl 125, NaHCO$_3$ 25, glucose 20, KCl 2.5, CaCl$_2$ 2, NaH$_2$PO$_4$ 1.25, MgCl$_2$1 (310 mOsm, pH 7.3 when bubbled with Carbogen [5% (vol/vol) O$_2$/95% (vol/vol) CO$_2$]). All recordings were restricted to lobules IV/V of the cerebellar vermis to reduce potential functional heterogeneity among different lobules (*Straub et al., 2020*).

## Presynaptic recordings and flash photolysis

All recordings were performed at near-physiological temperature by adjusting the set temperature of the TC-324B perfusion heat controller (Warner Instruments, Hamden, CT, United States) until the temperature in the center of the recording chamber with immersed objective was between 36.0°C and 36.3°C. This process was repeated before using a new brain slice. During recordings, the thermometer was put at the side of the recording chamber and the readout was monitored to avoid potential drifts in temperature (the readout was between 32°C and 34°C, critically depending on the position of the thermometer, and changed during recording from one brain slice by less than 0.5°C). The room temperature was controlled using an air conditioner set to 24°C. Presynaptic patch-pipettes were from pulled borosilicate glass (2.0/1.0 mm outer/inner diameter; Science Products) to open-tip resistances of 3–5 MΩ (when filled with intracellular solution) using a DMZ Puller (Zeitz-Instruments, Munich, Germany). Slices were superfused with artificial cerebrospinal fluid (ACSF) containing (in mM): NaCl 105, NaHCO$_3$ 25, glucose 25, TEA 20, 4-AP 5, KCl 2.5, CaCl$_2$ 2, NaH$_2$PO4 1.25, MgCl$_2$ 1, and tetrodotoxin (TTX) 0.001, equilibrated with 95% O$_2$ and 5% CO$_2$. Cerebellar mossy fiber boutons (cMFBs) were visualized with oblique illumination and infrared optics (*Ritzau-Jost et al., 2014*). Whole-cell patch-clamp recordings of cMFBs were performed using a HEKA EPC10/2 amplifier controlled by Patchmaster software (HEKA Elektronik, Lambrecht, Germany). The intracellular solution contained (in mM): CsCl 130, MgCl$_2$ 0.5, TEA-Cl 20, HEPES 20, Na$_2$ATP 5, NaGTP 0.3. For Ca$^{2+}$ uncaging experiments, equal concentrations of DM-nitrophen (DMn) and CaCl$_2$ were added depending on the aimed post-flash Ca$^{2+}$ concentration, such that either 0.5, 2, or 10 mM was used for low, middle, or high target range of post-flash Ca$^{2+}$ concentration, respectively (*Table 1*). To quantify post-flash Ca$^{2+}$ concentration with a previously established dual indicator method (see below; *Delvendahl et al., 2015*; *Sabatini et al., 2002*), Atto594, OGB-5N, and Fluo-5F were used at concentrations as shown in (*Table 1*).

A 50 mM solution stock of DMn was prepared by neutralizing 50 mM DMn in H$_2$O with 200 mM CsOH in H$_2$O. The purity of each DMn batch was determined in the intracellular solution used for patching through titration with sequential addition of Ca$^{2+}$ as previously described (*Schneggenburger, 2005*) and by measuring the Ca$^{2+}$ concentration using the dual indicator method with 10 µM Atto594 and 50 µM OGB1 (*Delvendahl et al., 2015*).

After waiting for at least one minute in whole-cell mode to homogenously load the terminal with intracellular solution, capacitance measurements were performed at a holding potential of −100 mV with sine-wave stimulation (5 kHz or 10 kHz frequency and ±50 mV amplitude; *Hallermann et al., 2003*). Distant mossy fiber boutons on the same axon are unlikely to contaminate capacitance measurements because investigations at hippocampal mossy fiber buttons indicate that the high-frequency sine-wave techniques as used in our study are hardly affected by release from neighboring boutons on the same axon (*Hallermann et al., 2003*). During the ongoing sine-wave stimulation, a UV laser source (375 nm, 200 mW, Rapp OptoElectronic) was used to illuminate the whole presynaptic terminal. According to a critical illumination, the end of the light guide of the UV laser was imaged into the focal plan resulting in a homogeneous illumination in a circular area of ~30 µm diameter (*Figure 2—figure supplement 1*). The duration of the UV illumination was 100 µs controlled with sub-microsecond precision by external triggering of the laser source. In capacitance

measurements with 10 kHz sine wave frequency, longer pulses of 200 µs were used to reach high $Ca^{2+}$ levels. In a subset of experiments, UV pulses of 1 ms were used to rule out fast undetectable $Ca^{2+}$ overshoots (*Bollmann et al., 2000*; *Figure 3—figure supplement 2*). The UV flash intensity was set to 100% and reduced in some experiments (10–100%) to obtain small elevations in $Ca^{2+}$ concentrations (*Table 1*). To avoid photoproducts-induced cell toxicity, we applied only one flash per recording. In a subset of the paired recordings with weak UV illumination (post-flash $Ca^{2+}$ concentration < 5 µM), we used consecutive flashes on the same cell (from 43 paired cells, 16 consecutive recordings were used).

## Paired recordings between cMFBs and GCs

For paired pre- and postsynaptic recordings, granule cells (GCs) were whole-cell voltage-clamped with intracellular solution containing the following (in mM): K-gluconate 150, NaCl 10, K-HEPES 10, MgATP three and Na-GTP 0.3 (300–305 mOsm, pH adjusted to 7.3 with KOH). 10 µM Atto594 was included to visualize the dendrites of the GCs (*Ritzau-Jost et al., 2014*). After waiting sufficient time to allow for the loading of the dye, the GC dendritic claws were visualized through two-photon microscopy, and subsequently, cMFBs near the dendrites were identified by infrared oblique illumination and were patched and loaded with caged $Ca^{2+}$ and fluorescent indicators as previously described. The reliable induction of an EPSC in the GC was used to unequivocally confirm a cMFB-GC synaptic connection. In a subset of the $Ca^{2+}$ uncaging experiments, simultaneous presynaptic capacitance and postsynaptic EPSC recordings were performed from cMFBs and GCs, respectively.

## Clamping intracellular basal $Ca^{2+}$ concentrations

The intracellular solution for presynaptic recordings of the data shown in *Figure 1* contained the following in mM: K-gluconate 150, NaCl 10, K-HEPES 10, MgATP 3, Na-GTP 0.3. With a combination of EGTA and $CaCl_2$ (5 mM EGTA / 0.412 mM $CaCl_2$ or 6.24 mM EGTA / 1.65 mM $CaCl_2$), we aimed to clamp the free $Ca^{2+}$ concentration to low and high resting $Ca^{2+}$ concentrations of ~50 or ~200 nM, respectively, while maintaining a free EGTA concentration constant at 4.47 mM. The underlying calculations were based on a $Ca^{2+}$ affinity of EGTA of 543 nM (*Lin et al., 2017*). The resulting free $Ca^{2+}$ concentration was quantified with the dual indicator method (see below) and was found to be to ~30 or ~180 nM, respectively (*Figure 1A*).

## Quantitative two-photon $Ca^{2+}$ imaging

For the quantification of $Ca^{2+}$ signals elicited through UV-illumination-induced uncaging, two-photon $Ca^{2+}$ imaging was performed as previously described (*Delvendahl et al., 2015*) using a Femto2D laser-scanning microscope (Femtonics) equipped with a pulsed Ti:Sapphire laser (MaiTai, Spectra-Physics) adjusted to 810 nm, a 60×/1.0 NA objective (Olympus), and a 1.4 NA oil-immersion condenser (Olympus). Data were acquired by doing line-scans through the cMFB. To correct for the flash-evoked luminescence from the optics, the average of the fluorescence from the line-scan in an area outside of the bouton was subtracted from the average of the fluorescence within the bouton (*Figure 2B*). Imaging data were acquired and processed using MES software (Femtonics). Upon releasing $Ca^{2+}$ from the cage, we measured the increase in the green fluorescence signal of the $Ca^{2+}$-sensitive indicator (OGB-5N or Fluo-5F) and divided it by the fluorescence of the $Ca^{2+}$-insensitive Atto594 (red signal). The ratio (R) of green-over-red fluorescence was translated into a $Ca^{2+}$ concentration through the following calculation (*Yasuda et al., 2004*).

$$[Ca^{2+}] = K_D \frac{(R - R_{min})}{(R_{max} - R)}$$

To avoid pipetting irregularities, which might influence the quantification of the fluorescence signals, pre-stocks of $Ca^{2+}$-sensitive and $Ca^{2+}$-insensitive indicators were used. For each pre-stock and each intracellular solution, 10 mM EGTA or 10 mM $CaCl_2$ were added to measure minimum ($R_{min}$) and maximum ($R_{max}$) fluorescence ratios, respectively. We performed these measurements in cMFBs and GCs as well as in cuvettes. Consistent with a previous report (*Delvendahl et al., 2015*), both $R_{min}$ and $R_{max}$ were higher when measured in cells than in cuvettes (by a factor of 1.73 ± 0.05; n = 83 and 63 measurements in situ and in cuvette; *Figure 3—figure supplement 3A*). The values in cMFBs and GCs were similar (*Figure 3—figure supplement 3B*). OGB-5N is not sensitive in detecting $Ca^{2+}$

concentrations less than 1 µM. Therefore, we deliberately adjusted $R_{min}$ of OGB-5N in the recordings where the pre-flash $Ca^{2+}$ had negative values, to a value resulting in a pre-flash $Ca^{2+}$ concentration of 60 nM, which corresponds to the average resting $Ca^{2+}$ concentration in these boutons (*Delvendahl et al., 2015*). This adjustment of $R_{min}$ resulted in a reduction of post-flash $Ca^{2+}$ amplitudes of on average $7.5 \pm 0.4\%$ (n = 37).

The fluorescence properties of DMn change after flash photolysis, and the $Ca^{2+}$ sensitive and insensitive dyes can differentially bleach during UV flash (*Schneggenburger, 2005*; *Zucker, 1992*). We assumed no effect of the UV flash on the $K_D$ of the $Ca^{2+}$-sensitive dyes (*Escobar et al., 1997*), and measured $R_{min}$ and $R_{max}$ before and after the flash for each used UV flash intensity and duration in each of the three solutions (*Table 1*; *Schneggenburger and Neher, 2000*). The flash-induced change was strongest for $R_{max}$ of solutions with OGB-5N, but reached only ~20% with the strongest flashes (*Figure 3—figure supplement 3F*).

## Deconvolution

Deconvolution of postsynaptic currents was performed essentially as described by *Ritzau-Jost et al., 2014*, based on routines developed by *Sakaba and Neher, 2001b*. The principle of this method is that the EPSC comprises currents induced by synchronous release and residual glutamate in the synaptic cleft due to delayed glutamate clearance and glutamate spill-over from neighboring synapses, which is prominent at the cMFB to GC synapses (*DiGregorio et al., 2002*). Kynurenic acid (2 mM) and Cyclothiazide (100 µM) were added to the extracellular solution to reduce postsynaptic receptor saturation and desensitization, respectively. The amplitude of the miniature EPSC (mEPSC) was set to the mean value of 10.1 pA ($10.1 \pm 0.2$ pA; n = 8) as measured in 2 mM kynurenic acid and 100 µM cyclothiazide. Kynurenic acid has been reported to absorb UV light resulting in a reduction of the uncaging efficiency (*Sakaba et al., 2005*; *Wölfel et al., 2007*). However, kynurenic acid particularly absorbs UV light at wavelength below 370 nm (*Wölfel et al., 2007*) suggesting that the reduction in the uncaging efficiency at the wavelength used in this study (375 nm) might be small. In agreement with this, we were able to increase the post-flash $Ca^{2+}$ concentrations to ~50 µM.

The deconvolution kernel had the following free parameters: the mEPSC early slope $\tau_0$, the fractional amplitude of the slow mEPSC decay phase $\alpha$, the time constant of the slow component of the decay $\tau_2$ of the mEPSC, the residual current weighting factor $\beta$, and the diffusional coefficient $d$. Applying the 'fitting protocol' described by *Sakaba and Neher, 2001b* before flash experiments might affect the number of vesicles released by subsequent $Ca^{2+}$ uncaging. On the other hand, applying the 'fitting protocol' after $Ca^{2+}$ uncaging might underestimate the measured number of vesicles due to flash-induced toxicity and synaptic fatigue especially when applying strong $Ca^{2+}$ uncaging. Therefore, we used the experiments with weak and strong flashes to extract the mini-parameters and the parameters for the residual current of the deconvolution kernel, respectively, as described in the following in more detail. To obtain the mini parameters (early slope, $\alpha$, and $\tau_2$) using weak flashes, deconvolution was first performed with a set of trial parameters for each cell pair. The mini-parameters of the deconvolution were optimized in each individual recording to yield low (but non-negative) step-like elevations in the cumulative release corresponding to small EPSCs measured from the postsynaptic terminal (the parameters for the residual current had little impact on the early phase of the cumulative release rate within the first 5 ms, therefore, some reasonable default values for the parameters of the residual current were used while iteratively adjusting the fast mini parameters for each individual recording). Next, using the average of the mini-parameters obtained from weak flashes, the deconvolution parameters for the residual current ($\beta$ and $d$) were optimized in each recording with strong flashes until no drops occurred in the cumulative release in the range of 5–50 ms after the stimulus (while iteratively readjusting the mini parameters, if needed, to avoid any drops in the cumulative release in the window of 5–10 ms that might arise when adjusting the slow parameters based on the cumulative release in the range of 5–50 ms). Finally, we averaged the values of each parameter and the deconvolution analysis of all recordings was re-done using the average parameters values. To test the validity of this approach, cumulative release from deconvolution of EPSCs and presynaptic capacitance recordings were compared in a subset of paired recordings (n = 9 pairs) similarly as done in previous investigations (*Ritzau-Jost et al., 2014*). Exponential fits to the cumulative release and the presynaptic capacitance traces provided very similar time constants. On a paired-wise comparison, the difference in the time constant was always less than 40%

(*Figure 3—figure supplement 4*). Therefore, both approaches yielded similar exponential time constants.

To combine the sustained release rate estimated from capacitance measurements (*Figure 5B*) and deconvolution analysis of EPSC (*Figure 5C*) for the modeling with release schemes (*Figure 6E*), we estimated the number of GCs per cMFB by comparing the product of the amplitude and the inverse of the time constant of the exponential fit of the presynaptic capacitance trace and the simultaneously measured cumulative release trace obtained by deconvolution analysis. Assuming a capacitance of 70 aF per vesicle (*Hallermann et al., 2003*), we obtained an average value of 90.1 GCs per MFB in close agreement with previous estimates using a similar approach (*Ritzau-Jost et al., 2014*). This connectivity ratio is larger than previous estimates (~10, *Billings et al., 2014*; ~50, *Jakab and Hamori, 1988*) which could be due to a bias toward larger terminals, ectopic vesicle release, postsynaptic rundown, or release onto Golgi cells.

## Measurement of $Ca^{2+}$ concentration using a $Ca^{2+}$-sensitive electrode

A precise estimation of the binding affinity of the $Ca^{2+}$-sensitive dyes is critical in translating the fluorescence signals into $Ca^2$ concentration. It has been reported that the $K_D$ of fluorescent indicators differs significantly depending on the solution in which it is measured (*Tran et al., 2018*) due to potential differences in ionic strength, pH, and concentration of other cations. Accordingly, different studies have reported different estimates of the $K_D$ of OGB-5N having an up to eight-fold variability (*Delvendahl et al., 2015*; *DiGregorio and Vergara, 1997*; *Neef et al., 2018*). In these studies, the estimation of the $K_D$ of the $Ca^{2+}$ sensitive dyes depended on the estimated $K_D$ of the used $Ca^{2+}$ chelator, which differs based on the ionic strength, pH, and temperature of the solution used for calibration. So, we set out to measure the $K_D$ of OGB-5N, in the exact solution and temperature, which we used during patching, through direct potentiometry using an ion-selective electrode combined with two-photon $Ca^{2+}$ imaging. An ion-selective electrode for $Ca^{2+}$ ions provides a direct readout of the free $Ca^{2+}$ concentration independent of the $K_D$ of the used $Ca^{2+}$ chelator. Using the same intracellular solution and temperature as used during experiments, the potential difference between the $Ca^{2+}$-sensitive electrode (ELIT 8041 PVC membrane, NICO 2000) and a single junction silver chloride reference electrode (ELIT 001 n, NIC0 2000) was read out with a pH meter in voltage mode. A series of standard solutions, with defined $Ca^{2+}$ concentration (Thermo Fisher Scientific) covering the whole range of our samples, were used to plot a calibration curve of the potential (mV) versus $Ca^{2+}$ concentration (µM). Then, the potential of several sample solutions containing the same intracellular solution used for patching, but with different $Ca^{2+}$ concentrations buffered with EGTA, was determined. This way, we got a direct measure of the free $Ca^{2+}$ concentration of several sample solutions, which were later used after the addition of $Ca^{2+}$-sensitive fluorometric indicators to plot the fluorescence signal of each solution versus the corresponding free $Ca^{2+}$ concentration verified by the $Ca^{2+}$-sensitive electrode, and accordingly the $K_D$ of the $Ca^{2+}$ indicators were obtained from fits with a Hill equation. The estimated $K_D$ was two-fold higher than the estimate obtained using only the $Ca^{2+}$ Calibration Buffer Kit (Thermo Fisher Scientific) without including intracellular patching solution (*Figure 3—figure supplement 1*). Comparable results were obtained when estimating the free $Ca^{2+}$ concentration using Maxchelator software (https://somapp.ucdmc.ucdavis.edu/pharmacology/bers/maxchelator/). Therefore, we used two independent approaches to confirm the $K_D$ of OGB-5N. We found that TEA increased the potential of the solutions measured through the $Ca^{2+}$-sensitive electrode, which is consistent with a previous report showing a similar effect of quaternary ammonium ions on potassium sensitive microelectrodes (*Neher and Lux, 1973*). We compared the fluorescence signals of our samples with or without TEA, to check if this effect of TEA is due to an interaction with the electrode or due to an effect on the free $Ca^{2+}$ concentration, and found no difference. Therefore, TEA had an effect on the electrode read-out without affecting the free $Ca^{2+}$, and accordingly, TEA was removed during the potentiometric measurements (*Figure 3—figure supplement 1*). This resulted in a good agreement of the estimates of the free $Ca^{2+}$ concentration measured using a $Ca^{2+}$-sensitive electrode and those calculated via Maxchelator.

## Assessment of the UV energy profile

The homogeneity of the UV laser illumination at the specimen plane was assessed in vitro by uncaging fluorescein (CMNB-caged fluorescein, Thermo Fisher Scientific). Caged fluorescein (2 mM) was

mixed with glycerol (5% caged fluorescein/ 95% glycerol) to limit the mobility of the released dye (*Bollmann et al., 2000*). We did the measurements at the same plane as we put the slice during an experiment. The fluorescence profile of the dye after being released from the cage was measured at different z-positions over a range of 20 µm. The intensity of fluorescein was homogenous over an area of 10 µm x 10 µm x 10 µm which encompasses the cMFB.

## Data analysis

The increase in membrane capacitance and in cumulative release based on deconvolution analysis was fitted with the following single or bi-exponential functions using Igor Pro (WaveMetrics) including a baseline and a variable onset.

$$f_{mono}(t) = \begin{cases} 0 & \text{if } t < d, \\ a\left(1 - exp\left[-\frac{(t-d)}{\tau}\right]\right) & \text{if } t \geq d \end{cases}$$

$$f_{bi}(t) = \begin{cases} 0 & \text{if } t < d, \\ a\left(1 - a_1 exp\left[-\frac{(t-d)}{\tau_1}\right] - (1-a_1) exp\left[-\frac{(t-d)}{\tau_2}\right]\right) & \text{if } t \geq d \end{cases} \tag{1}$$

where $d$ defines the delay, $a$ the amplitude, $\tau$ the time constant of the mono-exponetial fit, $\tau_1$ and $\tau_2$ the time constants of the fast and slow components of the bi-exponential fit, respectively, and $a_1$ the relative contribution of the fast component of the bi-exponential fit. The fitting of the release traces was always done with a time window of 5 ms before and 10 ms after flash onset. If the time constant of the mono-exponential fit exceeded 10 ms, a longer fitting duration of 60 ms after flash onset was used for both the experimental and the simulated data.

The acceptance of a bi-exponential fit was based on the fulfillment of the following three criteria: (1) at least 4% decrease in the sum of squared differences between the experimental trace and the fit compared with a mono-exponential fit ($\chi^2_{mono}/\chi^2_{bi} > 1.04$), (2) the time constants of the fast and the slow components differed by a factor >3, and (3) the relative contribution of each component was >10% (i.e. $0.1 < a_1 < 0.9$). If any of these criteria was not met, a mono-exponential function was used instead. In the case of weak flashes, where we could observe single quantal events within the initial part of the EPSC, mono-exponential fits were applied. In *Figure 1*, bi-exponential functions were used to fit the decay of the EPSC and the amplitude-weighted time constants were used (*Hallermann et al., 2010*).

Hill equations were used to fit the release rate versus intracellular $Ca^{2+}$ concentration on a double logarithmic plot according to the following equation:

$$H(x) = Log\left[V_{max}\frac{1}{1 + \left(\frac{K_D}{10^x}\right)^n}\right] \tag{2}$$

where $Log$ is the decadic logarithm, $V_{max}$ the maximal release rate, $K_D$ the $Ca^{2+}$ concentration at the half-maximal release rate, and $n$ the Hill coefficient. H(x) was fit on the decadic logarithm of the release rates and x was the decadic logarithm of the intracellular $Ca^{2+}$ concentration.

## Modeling of intra-bouton $Ca^{2+}$ dynamics

We simulated the intra-bouton $Ca^{2+}$ dynamics using a single compartment model. The kinetic reaction schemes for $Ca^{2+}$ and $Mg^{2+}$-uncaging and -binding (*Figure 6A*) were converted to a system of ordinary differential equations (ODEs) that was numerically solved using the NDSolve function in Mathematica 12 (Wolfram) as described previously (*Bornschein et al., 2019*). The initial conditions for the uncaging simulation were derived by first solving the system of ODEs for the steady state using total concentrations of all species and the experimentally determined $[Ca^{2+}]_{rest}$ as starting values. Subsequently, the values obtained for all free and bound species were used as initial conditions for the uncaging simulation. The kinetic properties of DMn were simulated according to *Faas et al., 2005*, *Faas et al., 2007*. The total DMn concentration ($[DMn]_T$) includes the free form ($[DMn]$), the $Ca^{2+}$-bound form ($[CaDMn]$), and the $Mg^{2+}$-bound form ($[MgDMn]$). Each of these forms is subdivided into an uncaging fraction (α) and a non-uncaging fraction (1-α). The uncaging fractions were further subdivided into a fast (af) and a slow (1-af) uncaging fraction:

$[DMn]_T = [DMn]_f + [DMn]_s + [CaDMn]_f + [CaDMn]_s + [MgDMn]_f + [MgDMn]_s$

$$[DMn] = [DMn]_f + [DMn]_s$$
$$[DMn]_f = \alpha\ af\ [DMn]$$
$$[DMn]_s = \alpha\ (1\text{-}af)\ [DMn]$$
$$[CaDMn] = [CaDMn]_f + [CaDMn]_s$$
$$[CaDMn]_f = \alpha\ af\ [CaDMn]$$
$$[CaDMn]_s = \alpha\ (1\text{-}af)\ [CaDMn]$$
$$[MgDMn] = [MgDMn]_f + [MgDMn]_s$$
$$[MgDMn]_f = \alpha\ af\ [MgDMn]$$
$$[MgDMn]_s = \alpha\ (1\text{-}af)\ [MgDMn]$$

The suffixes 'T', 'f', and 's' indicate total, fast or slow, respectively. The transition of fast and slow uncaging fractions into low-affinity photoproducts (PP) occurred with fast ($\tau_f$) or slow ($\tau_s$) time constants, respectively. Free $Ca^{2+}$ or $Mg^{2+}$-bound DMn decomposed into two photoproducts (PP1, PP2) differing with respect to their binding kinetics. The binding kinetics of all species were governed by the corresponding forward ($k_{on}$) and backward ($k_{off}$) rate constants

$$\frac{d[CaDMn]}{dt}_x = k_{on}[Ca][DMn]_x - k_{off}[CaDMn]_x - \frac{[CaDMn]_x}{\tau_x} H(t - t_{flash})\quad x = f, s$$

$$\frac{d[MgDMn]}{dt}_x = k_{on}[Mg][DMn]_x - k_{off}[MgDMn]_x - \frac{[MgDMn]_x}{\tau_x} H(t - t_{flash})\quad x = f, s$$

$$\frac{d[DMn]}{dt}_x = -k_{on}[Ca][DMn]_x + k_{off}[CaDMn]_x - k_{on}[Mg][DMn]_x + k_{off}[MgDMn]_x$$
$$cc$$
$$-\frac{[DMn]_x}{\tau_x} H(t - t_{flash})\quad x = f, s$$

$$\frac{d[CaPP1]}{dt} = k_{on}[Ca][PP1] - k_{off}[CaPP1] + \frac{[CaDMn]_f}{\tau_f} H(t - t_{flash}) + \frac{[CaDMn]_s}{\tau_s} H(t - t_{flash})$$

$$\frac{d[MgPP1]}{dt} = k_{on}[Mg][PP1] - k_{off}[MgPP1]$$

$$\frac{d[PP1]}{dt} = -k_{on}[Ca][PP1] + k_{off}[CaPP1] - k_{on}[Mg][PP1] + k_{off}[MgPP1]$$
$$+ \frac{[CaDMn]_f}{\tau_f} H(t - t_{flash}) + \frac{[CaDMn]_s}{\tau_s} H(t - t_{flash})$$

$$\frac{d[CaPP2]}{dt} = k_{on}[Ca][PP2] - k_{off}[CaPP2]$$

$$\frac{d[MgPP2]}{dt} = k_{on}[Mg][PP2] - k_{off}[MgPP2] + \frac{[MgDMn]_f}{\tau_f} H(t - t_{flash})$$
$$+ \frac{[MgDMn]_s}{\tau_s} H(t - t_{flash})$$

$$\frac{d[PP2]}{dt} = -k_{on}[Ca][PP2] + k_{off}[CaPP2] - k_{on}[Mg][PP2] + k_{off}[MgPP2]$$

$$+ 2\ \frac{[DMn]_f}{\tau_f} H(t - t_{flash}) + \frac{[DMn]_s}{\tau_s} H(t - t_{flash})$$

$$+ \frac{[MgDMn]_f}{\tau_f} H(t - t_{flash}) + \frac{[MgDMn]_s}{\tau_s} H(t - t_{flash})$$

where $H$ is the Heaviside step function and $t_{flash}$ the time of the UV flash. $Ca^{2+}$ and $Mg^{2+}$-binding to the dye, ATP, and an endogenous buffer (EB) were simulated by second-order kinetics:

$$\frac{d[Ca]}{dt}_{buffer} = -k_{on,j}[Ca][B] + k_{off,j}[CaB]\quad j = dye, ATP, EB$$

$$\frac{d[Mg]}{dt} = -k_{on,j}[Mg][B] + k_{off,j}[MgB] \quad j = ATP$$

$$\frac{d[B]}{dt} = -\frac{d[CaB]}{dt} - \frac{d[MgB]}{dt} \quad B = dye, ATP, EB$$

The time course of the total change in $Ca^{2+}$ concentration or $Mg^{2+}$ concentration is given by the sum of all the above equations involving changes in $Ca^{2+}$ concentration or $Mg^{2+}$ concentration, respectively. $Ca^{2+}$ concentration as reported by the dye was calculated from the concentration of the $Ca^{2+}$-dye complex assuming equilibrium conditions (*Markram et al., 1998*). The clearing of $Ca^{2+}$ from the cytosol was not implemented in these simulations. Instead, the $Ca^{2+}$ concentration was simulated only for 10 ms after the flash. The experimentally observed subsequent decay of the $Ca^{2+}$ concentration was implemented by an exponential decay to the resting $Ca^{2+}$ concentration with a time constant of 400 ms. The parameters of the model are given in *Table 2*.

These simulations were used to obtain $Ca^{2+}$ transients with peak amplitudes covering the entire range of post-flash $Ca^{2+}$ concentrations. To this end, the uncaging efficiency α was varied in each of the three experimentally used combinations of concentrations of DMn and $Ca^{2+}$ indicators (see *Table 1* for details).

## Modeling of release schemes

Model one with two $Ca^{2+}$ binding steps mediating fusion and one $Ca^{2+}$-dependent priming step was defined according to the following differential equation

$$\begin{pmatrix} dV_{0Ca}(t)/dt \\ dV_{1Ca}(t)/dt \\ dV_{2Ca}(t)/dt \\ dV_{fused}(t)/dt \end{pmatrix} = M \begin{pmatrix} V_{0Ca}(t) \\ V_{1Ca}(t) \\ V_{2Ca}(t) \\ V_{fused}(t) \end{pmatrix}$$

$V_{0Ca}$, $V_{1Ca}$, and $V_{2Ca}$ denote the fraction of vesicles with a fusion sensor with 0 to 2 bound $Ca^{2+}$ ions, respectively, and $V_{fused}$ denotes the fused vesicles as illustrated in *Figure 6D*. The reserve pool $V_R$ is considered to be infinite. $M$ denotes the following 4x4 matrix:

$$\begin{pmatrix} -2k_{on}-k_{unprim}+k_{prim}/V_{0Ca}(t) & k_{off} & 0 & 0 \\ 2k_{on} & -k_{off}-k_{on} & 2k_{off}b & 0 \\ 0 & k_{on} & -y-2k_{off}b & 0 \\ 0 & 0 & y & 0 \end{pmatrix}$$

See *Table 3* for the values and $Ca^{2+}$-dependence of the rate constants in the matrix.

The initial condition was defined as $V_{0Ca}(0) = k_{prim}/k_{unprim}$ and $V_{1Ca}(0)$, $V_{2Ca}(0)$, and $V_{fused}(0)$ was zero. $k_{prim}$ was the sum of a $Ca^{2+}$-dependent and $Ca^{2+}$-independent rate constants. The $Ca^{2+}$-dependence was implemented as a Michaelis-Menten kinetic with a maximum rate constant of 30 $s^{-1}$ (*Ritzau-Jost et al., 2014*) and a $K_D$ of 2 µM (*Miki et al., 2018*). The $Ca^{2+}$-independent rate constant was 0.6 $s^{-1}$, adjusted to reproduce the factor of 3 upon elevating $Ca^{2+}$ from 30 to 180 nM (*Figures 1D* and *7D*). $k_{unprim}$ was defined such that the occupancy $V_{0Ca}(0) = 1$ for the default pre-flash resting $Ca^{2+}$ concentration of 227 nM (*Tables 2* and *3*). The occupancy was set to 1 for simplicity in all models because our data did not allow determining the occupancy (but for evidence of occupancy < 1 see *Pulido and Marty, 2017*).

The differential equations were solved with the NDSolve function of Mathematica. The $Ca^{2+}$ concentration, $Ca^{2+}(t)$, was obtained from the simulations as described in the previous paragraph. $V_{fused}(t)$ represents the cumulative release normalized to the pool of release-ready vesicles per cMFB to GC connection. To reproduce the absolute sustained release rate (*Figures 5*, *6D*), $V_{fused}(t)$ was multiplied by a pool of release-ready vesicles per connection of 10 vesicles. The cumulative release, $V_{fused}(t)$, including a pre-flash baseline was sampled with 5 or 10 kHz. Realistic noise for 5- or 10-kHz-capacitance or deconvolution measurements was added and the data, in the 10 ms-window after the flash, were fit with mono- and bi- exponential functions (*Equation 1*). The selection of a bi- over a mono-exponential fit was based on identical criteria as in the analysis of the experimental data including the prolongation of the fitting duration from 10 to 60 ms if the time constant of the mono-exponential fit was >10 ms (see section Data analysis). For each peak post-flash $Ca^{2+}$ concentration

(i.e. simulated $Ca^{2+}(t)$ transient) the sampling, addition of noise, and exponential fitting were repeated 50 times. The median of these values represents the prediction of the model for each peak post flash $Ca^{2+}$ concentration. The parameters of the r were manually adjusted to obtain best-fit results.

Model 2 was a sequential two-pool model based on *Miki et al., 2018* with five $Ca^{2+}$ binding steps mediating fusion and two $Ca^{2+}$-dependent priming steps defined according to the following differential equations

$$
\begin{pmatrix}
dV_{2,0Ca}(t)/dt \\
dV_{2,1Ca}(t)/dt \\
dV_{2,2Ca}(t)/dt \\
dV_{2,3Ca}(t)/dt \\
dV_{2,4a}(t)/dt \\
dV_{2,5Ca}(t)/dt \\
dV_{2,fused}(t)/dt
\end{pmatrix}
= M
\begin{pmatrix}
V_{2,0Ca}(t) \\
V_{2,1Ca}(t) \\
V_{2,2Ca}(t) \\
V_{2,3Ca}(t) \\
V_{2,4Ca}(t) \\
V_{2,5Ca}(t) \\
V_{2,fused}(t)
\end{pmatrix}
$$

$V_{2,0Ca}$, $V_{2,1Ca}$, ..., and $V_{2,5Ca}$ denote the fraction of vesicles with a fusion sensor with 0 to 5 bound $Ca^{2+}$ ions, respectively, and $V_{2,fused}$ denotes fused vesicles as illustrated in *Figure 6D*. The fraction of vesicles in state $V_1$ is calculated according to the following differential equation

$$
\frac{dV_1(t)}{dt} = k_{prim1} - k_{unprim1} \, V_1(t) - k_{prim2} \, V_1(t) + k_{unprim2} \, V_{2,0Ca}(t)
$$

$M$ denotes the following 7x7 matrix:

$$
\begin{pmatrix}
-5k_{on} - k_{unprim2} + k_{prim2}/V_1(t)/V_{2,0Ca}(t) & k_{off} & 0 & 0 & 0 & 0 & 0 \\
5k_{on} & -k_{off} - 4k_{on} & 2k_{off}b & 0 & 0 & 0 & 0 \\
0 & 4k_{on} & -2k_{off}b - 3k_{on} & 3k_{off}b^2 & 0 & 0 & 0 \\
0 & 0 & 3k_{on} & -3k_{off}b^2 - 2k_{on} & 4k_{off}b^3 & 0 & 0 \\
0 & 0 & 0 & 2k_{on} & -4k_{off}b^3 - k_{on} & 5k_{off}b4 & 0 \\
0 & 0 & 0 & 0 & k_{on} & -y - 5k_{off}b^4 & 0 \\
0 & 0 & 0 & 0 & 0 & y & 0
\end{pmatrix}
$$

To implement the use-dependent slowing of the release rate constants of this model (*Miki et al., 2018*) in a deterministic way, a site-plugging state, $P(t)$, was defined according to

$$
\frac{dP(t)}{dt} = (1 - P(t)) \frac{dV_{2,fused}}{dt}(t) - 40ms \, P(t) \tag{3}
$$

$P(t)$ is approaching one during strong release and decays with a time constant of 40 ms back to zero. Similar to the implementation by *Miki et al., 2018*, the rate constants $k_{on}$ and $k_{off}$ were linearly interpolated between two values depending on $P(t)$ as

$$
k_{on}(t) = k_{on,init} + (k_{on,plugged} - k_{on,init}) \, P(t) \tag{4}
$$

$$
k_{off}(t) = k_{off,init} + (k_{off,plugged} - k_{off,init}) \, P(t)
$$

The reserve pool $V_R$ is considered to be infinite. See *Table 3* for the values and $Ca^{2+}$-dependence of the rate constants in these differential equations.

The initial condition is defined as $V_1(0) = k_{prim1}/k_{unprim1}$ and $V_{2,0Ca}(0) = (k_{prim1}/k_{unprim1})*(k_{prim2}/k_{unprim2})$. The initial condition of the other state $V_{2,1Ca}(0)$ to $V_{5,0Ca}(0)$, $V_{fused}(0)$, and $P(0)$ were zero. $k_{prim1}$ and $k_{prim2}$ were the sum of a $Ca^{2+}$-dependent and $Ca^{2+}$-independent rate constant defined similarly as described in *Miki et al., 2018* and adjusted as described for model 1. $k_{unprim1}$ and $k_{unprim2}$ were defined such that the occupancy $V_1(0) = 1$ and $V_{2,0Ca}(0) = 1$ for the default pre-flash resting $Ca^{2+}$ concentration of 227 nM (*Tables 2* and *3*).

Model 3 was a parallel two-pool model similar as described by *Voets, 2000* and *Walter et al., 2013* but with five $Ca^{2+}$ binding steps mediating fusion of both types of vesicles and a $Ca^{2+}$-independent priming step for $V_1$ vesicles and a $Ca^{2+}$-dependent transition step from $V_1$ to $V_2$ vesicles defined according to the following differential equations

$$
\begin{pmatrix}
dV_{1,0Ca}(t)/dt \\
dV_{1,1Ca}(t)/dt \\
dV_{1,2Ca}(t)/dt \\
dV_{1,3Ca}(t)/dt \\
dV_{1,4a}(t)/dt \\
dV_{1,5Ca}(t)/dt \\
dV_{1,fused}(t)/dt
\end{pmatrix}
= M_1
\begin{pmatrix}
V_{1,0Ca}(t) \\
V_{1,1Ca}(t) \\
V_{1,2Ca}(t) \\
V_{1,3Ca}(t) \\
V_{1,4Ca}(t) \\
V_{1,5Ca}(t) \\
V_{1,fused}(t)
\end{pmatrix}
$$

$$
\begin{pmatrix}
dV_{2,0Ca}(t)/dt \\
dV_{2,1Ca}(t)/dt \\
dV_{2,2Ca}(t)/dt \\
dV_{2,3Ca}(t)/dt \\
dV_{2,4a}(t)/dt \\
dV_{2,5Ca}(t)/dt \\
dV_{2,fused}(t)/dt
\end{pmatrix}
= M_2
\begin{pmatrix}
V_{2,0Ca}(t) \\
V_{2,1Ca}(t) \\
V_{2,2Ca}(t) \\
V_{2,3Ca}(t) \\
V_{2,4Ca}(t) \\
V_{2,5Ca}(t) \\
V_{2,fused}(t)
\end{pmatrix}
$$

$V_{1,0Ca}, V_{1,1Ca}, \ldots$ and $V_{1,5Ca}$ denote the fraction of vesicles with a low-affinity fusion sensor with 0 to 5 bound $Ca^{2+}$ ions, respectively, and $V_{2,0Ca}, V_{2,1Ca}, \ldots$ and $V_{2,5Ca}$ denote the fraction of vesicles with a high-affinity fusion sensor with 0 to 5 bound $Ca^{2+}$ ions, respectively. $V_{1,fused}$ and $V_{2,fused}$ denote fused vesicles as illustrated in **Figure 6D**.

$M_1$ denotes the following 7x7 matrix:

$$
\begin{pmatrix}
-5k_{on1}-k_{unprim1}-k_{prim2}+k_{prim1}/V_{1,0Ca}(t)+k_{unprim2}/V_{2,0Ca}(t)/V_{1,0Ca}(t) & k_{off1} & 0 & 0 & 0 & 0 & 0 \\
5k_{on1} & -k_{off1}-4k_{on1} & 2k_{off1}b & 0 & 0 & 0 & 0 \\
0 & 4k_{on1} & -2k_{off1}b-3k_{on1} & 3k_{off1}b^2 & 0 & 0 & 0 \\
0 & 0 & 3k_{on1} & -3k_{off1}b^2-2k_{on1} & 4k_{off1}b^3 & 0 & 0 \\
0 & 0 & 0 & 2k_{on1} & -4k_{off1}b^3-k_{on1} & 5k_{off1}b4 & 0 \\
0 & 0 & 0 & 0 & k_{on1} & -y-5k_{off1}b4 & 0 \\
0 & 0 & 0 & 0 & 0 & y & 0
\end{pmatrix}
$$

$M_2$ denotes the following 7x7 matrix:

$$
\begin{pmatrix}
-5k_{on1}-k_{unprim2}+k_{prim2}V_{1,0Ca}(t)/V_{2,0Ca}(t) & k_{off2} & 0 & 0 & 0 & 0 & 0 \\
5k_{on2} & -k_{off2}-4k_{on2} & 2k_{off2}b & 0 & 0 & 0 & 0 \\
0 & 4k_{on2} & -2k_{off2}b-3k_{on2} & 3k_{off2}b^2 & 0 & 0 & 0 \\
0 & 0 & 3k_{on2} & -3k_{off2}b^2-2k_{on2} & 4k_{off2}b^3 & 0 & 0 \\
0 & 0 & 0 & 2k_{on2} & -4k_{off2}b^3-k_{on2} & 5k_{off2}b4 & 0 \\
0 & 0 & 0 & 0 & k_{on2} & -y-5k_{off2}b4 & 0 \\
0 & 0 & 0 & 0 & 0 & y & 0
\end{pmatrix}
$$

The initial condition is defined as $V_{2,0Ca}(0) = k_{prim1}/k_{unprim1}$ and $V_{2,0Ca}(0) = (k_{prim1}/k_{unprim1})*(k_{prim2}/k_{unprim2})$. The initial condition of the other state $V_{1,1Ca}(0)$ to $V_{1,0Ca}(0)$, $V_{1,fused}(0)$, and $V_{2,1Ca}(0)$ to $V_{2,0Ca}(0)$, $V_{2,fused}(0)$ were zero. $k_{prim1}$ was a $Ca^{2+}$-independent rate constant and $k_{prim2}$ was the sum of a $Ca^{2+}$-dependent and $Ca^{2+}$-independent rate constants defined similarly as described in *Hallermann et al., 2010* and adjusted as described for model 1. $k_{unprim1}$ and $k_{unprim2}$ were defined such that the occupancy $V_{1,0Ca}(0) = $ one and $V_{2,0Ca}(0) = $ one for the default pre-flash resting $Ca^{2+}$ concentration of 227 nM (*Tables 2* and *3*).

## Statistical analysis

Boxplots show median and 1st/3rd quartiles with whiskers indicating the whole data range (*Figures 1* and *7*). For statistical comparison, Mann-Whitney *U* tests were used, and the p-values are indicated above the boxplots.

## Acknowledgements

We thank Erwin Neher for help with algorithms for calculating the $Ca^{2+}$ concentration of the intracellular solutions (Figure 1) and for helpful discussions. This work was supported by a European Research Council Consolidator Grant (ERC CoG 865634) to SH and by the German Research Foundation (DFG; SCHM1838/2-1) to HS and HA6386/10-1 to SH.

# Additional information

## Funding

| Funder | Grant reference number | Author |
| --- | --- | --- |
| Deutsche Forschungsge-meinschaft | SCHM1838/2-1 | Hartmut Schmidt |
| Deutsche Forschungsge-meinschaft | HA6386/10-1 | Stefan Hallermann |
| European Research Council | CoG 865634 | Stefan Hallermann |

The funders had no role in study design, data collection and interpretation, or the decision to submit the work for publication.

## Author contributions

Abdelmoneim Eshra, Conceptualization, Data curation, Investigation, Visualization, Methodology, Writing - original draft, Writing - review and editing; Hartmut Schmidt, Investigation, Visualization, Methodology, Writing - original draft, Writing - review and editing; Jens Eilers, Methodology, Writing - review and editing; Stefan Hallermann, Conceptualization, Software, Investigation, Methodology, Writing - original draft, Writing - review and editing

## Author ORCIDs

Abdelmoneim Eshra (iD) https://orcid.org/0000-0002-6579-3000
Hartmut Schmidt (iD) http://orcid.org/0000-0002-9516-423X
Stefan Hallermann (iD) https://orcid.org/0000-0001-9376-7048

## Ethics

Animal experimentation: All animals were treated in accordance with the German Protection of Animals Act and with the guidelines for the welfare of experimental animals issued by the European Communities Council Directive (EU Directive 2010/63/EU). The animal killing report number is T41/16 (gov. ID: DD24-5131/347/44).

## Decision letter and Author response

Decision letter https://doi.org/10.7554/eLife.70408.sa1
Author response https://doi.org/10.7554/eLife.70408.sa2

# Additional files

## Supplementary files

• Transparent reporting form

## Data availability

The code of the simulations of the release schemes (models 1 to 3) is available at https://github.com/HallermannLab/2021_eLife (copy archived at https://archive.softwareheritage.org/swh:1:rev:4cd31058945a8f1c8364d8fc21f0a2902de33365).

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
