## [Decision Letter]

**Acceptance summary:**

The authors examined the calcium dependence of exocytosis at cerebellar mossy fiber boutons using multiple experimental approaches - electrophysiology, calcium imaging, calcium uncaging and capacitance measurements. These specific approaches have been used at other synapses over many years, and the authors show intriguing differences among different types of synapses that are likely functionally significant. A strength of this work is the masterful implementation and explanation of the techniques that combine to lend the paper reliability and give it lasting value.

**Decision letter after peer review:**

Thank you for submitting your article "Calcium dependence of neurotransmitter release at a high fidelity synapse" for consideration by eLife. Your article has been reviewed by 3 peer reviewers, and the evaluation has been overseen by Gary Westbrook as the Senior and Reviewing Editor. The reviewers have opted to remain anonymous.

The reviewers have discussed their reviews with one another, and the Reviewing Editor has drafted the below summary to help you prepare a revised submission. For the most part, the revisions can be addressed by more better discussion of the issues enumerated below and by addressing limitations of the approach.

Essential revisions:

1. The introduction could be improved. The authors do a good job on the calcium dependence of release, but the presentation of priming and replenishment is confusing. They should set up the idea that the time course of release will be used to provide insight into replenishment. The issues of basal calcium on the amplitude of evoked responses is unlikely to be the same thing as replenishment (maybe they dont either, but they should spell this out better). Part of the problem is that they ignore a long and rich history of studies describing how residual calcium leads to enhancement of neurotransmitter release, even though the experiments examining how basal free calcium levels influence neurotransmitter release are very closely related to those papers. They finally (and briefly) discuss a small fraction of that literature in the discussion (and they could do a much better job of it).

2. MFs come from a variety of sources and are likely to have highly heterogeneous properties. This is not adequately dealt with in the current study. Chabrol et al ( 2015 Synaptic diversity enables temporal coding of coincident multisensory inputs in single neurons) previously studied the properties of MF synapses and found tremendous diversity in their release properties. The conclusion was that that the origin of MFs strongly influenced their release properties. Based on the properties they describe it would be reasonable to assume that the calcium dependence of release could be very different for MFs from different sources. That study was performed in lobule X, where there may be more diversity. The current study was performed in lobules IV and V, which reduces interlobule variability but not intralobule variability. I do not think that there has been a comparable study characterizing the diversity of MF inputs in these regions. They need to deal with this issue and the complications it would introduce into this study.

3. Comment not requiring additional experiments: Stefan Hallermann has previously published a beautiful paper describing the role of Bassoon in vesicle reloading at mossy fiber synapses (Hallermann, Neuron 2010). If they had studied knockout animals in the current study, they could make a much stronger case about vesicle replenishment, and it would have the potential to be a much more convincing paper.

4. 36C is not really a physiological temperature. Although there seems to be some variability in the physiological temperature it seems as if 37.5-38.5 is a better approximation of physiological temperature. See: Kiyatkin EA (2019) Brain temperature and its role in physiology and pathophysiology: Lessons from 20 years of thermo-recording. Also, they state that experiments were done at 36C. I think many experimenters find that there is some fluctuation in the actual temperature of the slice. The authors should provide an accurate estimate of the range of temperatures used in their experiment.

5. Does the elevation of basal calcium affect mEPSC frequency? It would be useful to know the number of claws on each recorded granule cell and scale mEPSC frequency effects accordingly in order to estimate the effects on release from the recorded mossy fiber. It is important to know whether the same elevation of basal calcium is sufficient to influence release in other ways.

6. Organization of the results . One reviewer wondered why the authors started with the basal calcium experiment? " I like it. I think it is interesting. But it seems slightly unrelated to much of the rest of the paper and is of a very different flavor. It also seems almost unrelated to the introduction. It is not very well integrated into the rest of the paper. It seems as if it would be better to move it to near Figure 7."

7. Distant mossy fibers on the same axon could complicate capacitance measurements. Have the authors considered how this could affect their capacitance measurements?

8. How might endocytosis influence the capacitance measurements?

9. On line 245-247 the authors say that they did simultaneous capacitance and deconvolution measurements. Please present these data in a supplementary figure to show the extent to which they agree and disagree. It would be useful to have a quantitative comparison of these two approaches. This is particularly important because it seems as if there is a big difference between the two approaches in 5B.

10. One reviewer raised a concern with Fig 5B. It seems as if the capacitance is measuring from the entire presynaptic bouton (release onto many granule cells) and the granule cell recording is extremely complicated because it has a direct component and a scaled down spillover component that is difficult to convert into vesicle number.

11. Please add additional discussion of the direct release and the spillover release components at mossy fiber synapses and how it could affect their measurements.

12. On page 32, it would seem that if you did simulations, you would expect an overshoot. Why isn't there an overshoot for brief uncaging?

13. The paper would great benefit from a more complete Ca2+ dose response curve. Because the authors goal is to completely characterize how release is regulated at the cerebellar mossy fiber bouton, it would significantly strengthen the conclusions of the manuscript to provide data on release rates at Ca2+ concentrations under 1 µM and higher than 50 µM in the dose response curve. We recognize that these experiments may not be completely feasible, but the issue should be openly discussed in the manuscript. The authors demonstrate that the release machinery is not saturated up to 50 µM Ca2+. It would be ideal to determine if the release machinery can be saturated.

[Editors' note: further revisions were suggested prior to acceptance, as described below.]

Thank you for resubmitting your work entitled "Calcium dependence of neurotransmitter release at a high fidelity synapse" for further consideration by eLife. Your revised article has been evaluated by Gary Westbrook (Senior Editor) . The manuscript has been improved but there are some remaining issues that need to be addressed, as outlined below:

As you will see the authors are pleased with the quality and rigor of the experiments. However, we really want you to address the issue raised by Reviewer 1 concerning the intellectual and experimental background that are critical to the eventual standing of this work.*Reviewer #1*:

This is an impressive paper. Its great strength is that it addresses an important question so carefully and thoroughly that it makes a substantial contribution to the field.

My one quibble it that I still think they have done a woefully inadequate job of discussing a great deal of previous literature centered around the residual calcium hypothesis. They have added a new section and an eclectic set of new references (55-80), but they disregard many classic papers and relevant reviews in the field. The residual calcium hypothesis has been most extensively discussed in terms of forms of short-term synaptic plasticity, such as facilitation, augmentation and post-tetanic potentiation. The literature goes about 70 years. I will leave it up to the authors to fix the issue to their satisfaction. Even though it is a minor point, I think this reflects badly on the authors. I leave it up to them to fix it to their own satisfaction.

*Reviewer #2:*

The authors have done a good job responding to the reviews. Their response to point 7 should have led to a comment (about the complications of distal terminals to Cm measurements) somewhere in the methods, rather than just a comment to the reviewer. Otherwise, I have no further concerns with the work and feel it is ready for publication. It is a fine paper.

---

## [Author Response]

Essential revisions:1. The introduction could be improved. The authors do a good job on the calcium dependence of release, but the presentation of priming and replenishment is confusing. They should set up the idea that the time course of release will be used to provide insight into replenishment. The issues of basal calcium on the amplitude of evoked responses is unlikely to be the same thing as replenishment (maybe they don’t either, but they should spell this out better). Part of the problem is that they ignore a long and rich history of studies describing how residual calcium leads to enhancement of neurotransmitter release, even though the experiments examining how basal free calcium levels influence neurotransmitter release are very closely related to those papers. They finally (and briefly) discuss a small fraction of that literature in the discussion (and they could do a much better job of it).

We thank the reviewers for this comment. We re-wrote the introduction and now discussed more literature on the impact of residual Ca^2+^ on neurotransmitter release in the introduction and discussion (for example line 55 to 80 of the introduction). Furthermore, we now explain already in the introduction how priming, fusion, and replenishment were dissected (line 93 to 99):

“To dissect the Ca^2+^-dependence of vesicle priming, we first directly manipulated the free basal intracellular Ca^2+^ concentration and measured the amount of action potential-evoked release. To determine the Ca^2+^-sensitivity of vesicle fusion, we measured the initial release kinetics of the fusion of the primed vesicles upon Ca^2+^ uncaging (with time constants mostly << 10 ms). To finally address the Ca^2+^-dependence of vesicle replenishment, we focused on the sustained component of release occurring during 100 ms of flash-evoked Ca^2+^ increase.”

2. MFs come from a variety of sources and are likely to have highly heterogeneous properties. This is not adequately dealt with in the current study. Chabrol et al ( 2015 Synaptic diversity enables temporal coding of coincident multisensory inputs in single neurons) previously studied the properties of MF synapses and found tremendous diversity in their release properties. The conclusion was that that the origin of MFs strongly influenced their release properties. Based on the properties they describe it would be reasonable to assume that the calcium dependence of release could be very different for MFs from different sources. That study was performed in lobule X, where there may be more diversity. The current study was performed in lobules IV and V, which reduces interlobule variability but not intralobule variability. I do not think that there has been a comparable study characterizing the diversity of MF inputs in these regions. They need to deal with this issue and the complications it would introduce into this study.

We thank the reviewer for this important comment. We agree that cMFBs could be functionally and molecularly distinct in lobule IV/V, similar to what has been reported in lobule IX in Chabrol et al. (Nat Neurosci, 2015). We now discussed this further as general caveat of this study in lines 604-612:

“The here obtained dose-response curve has the following three caveats. First, the cMFB to GC synapses in lobule IX are functionally distinct based on the origin of the mossy fibers (Chabrol et al., 2015). Therefore, the here-recorded boutons in lobule IV/V could be molecularly and functionally distinct leading to the observed scatter in the dose-response curve, which could cause an apparent shallowing. Yet, the scatter in the dose-response curve of comparable studies measuring the Ca^2+^ dependence of release at the calyx of Held synapse could in part also be due to the recently revealed heterogeneity between different classes of functionally and structurally distinct calyces (Grande and Wang, 2011).”

3. Comment not requiring additional experiments: Stefan Hallermann has previously published a beautiful paper describing the role of Bassoon in vesicle reloading at mossy fiber synapses (Hallermann, Neuron 2010). If they had studied knockout animals in the current study, they could make a much stronger case about vesicle replenishment, and it would have the potential to be a much more convincing paper.

We are relieved that the editor and the reviewers agreed that repeating the analysis on knockout mice would be beyond the scope of this manuscript. We would like to point out that this is the first study measuring the Ca^2+^ dependence of release at the cerebellar mossy fiber to granule cell synapse and combining UV Ca^2+^ uncaging and quantitative two-photon Ca^2+^ imaging with presynaptic capacitance measurements and deconvolution techniques during paired pre- and postsynaptic recordings. It was a great effort to establish the methods used in this manuscript, and to provide a detailed analysis which has been done at this level of detail only at the calyx of Held. We believe that this manuscript focusing only on wild-type cerebellar mossy fiber synapses provides relevant novel findings.

4. 36C is not really a physiological temperature. Although there seems to be some variability in the physiological temperature it seems as if 37.5-38.5 is a better approximation of physiological temperature. See: Kiyatkin EA (2019) Brain temperature and its role in physiology and pathophysiology: Lessons from 20 years of thermorecording. Also, they state that experiments were done at 36C. I think many experimenters find that there is some fluctuation in the actual temperature of the slice. The authors should provide an accurate estimate of the range of temperatures used in their experiment.

We thank the reviewer for pointing this out. We now explained in more details how we controlled the recording temperature in the methods section lines 742-750:

“All recordings were performed at near-physiological temperature by adjusting the set temperature of the TC-324B perfusion heat controller (Warner Instruments, Hamden, CT, United States) until the temperature in the center of the recording chamber with immersed objective was between 36.0 and 36.3°C. This process was repeated before using a new brain slice. During recordings, the thermometer was put at the side of the recording chamber and the readout was monitored to avoid potential drifts in temperature (the readout was between 32 to 34°C critically depending on the position of the thermometer and changed during recording of one brain slice by less than 0.5°C). The room temperature was controlled using an air conditioner set to 24°C.”

5. Does the elevation of basal calcium affect mEPSC frequency? It would be useful to know the number of claws on each recorded granule cell and scale mEPSC frequency effects accordingly in order to estimate the effects on release from the recorded mossy fiber. It is important to know whether the same elevation of basal calcium is sufficient to influence release in other ways.

We analyzed the mEPSC frequency in both conditions and found a trend towards higher mEPSC in the elevated basal Ca^2+^ experiments (see Author response image 1). In general, the frequency is higher compared to a previous publication (Hallermann et al., Neuron, 2010) and there is substantial cell-to-cell variability. This is most likely due to the fact that we unfortunately did not record longer periods at rest to analyze the mini frequency reliably. Instead, these miniature currents were analyzed in-between the current injections used to elicit an action potential as shown in Fig. 1B, which could artificially elevate the mini frequency. We unfortunately also did not acquire three-dimensional imaging data allowing to count the number of claws on each recorded granule cell. We therefore decided to not show these data in a figure in the manuscript but only to mention it in the following way (line 130-133):

“Interestingly, the frequency of miniature currents in-between the current injections used to elicit action potentials had a tendency to increase with elevated basal Ca^2+^ concentration (median 1.1 and 3.5 for the low and high basal Ca^2+^ conditions, respectively, n = 8 and 8; P_Mann-Whitney_ = 0.13; data not shown).”

**Author response image 1. respfig1:** Miniature EPSC frequency with 30 and 180 nM intracellular Ca^2+^ concentration. Boxplots show median and 1st/3rd quartiles with whiskers indicating the whole data range superimposed with the data from individual mossy fiber to granule cell pairs (n = 8 and 8 for the low and high basal Ca^2+^ conditions, respectively).

6. Organization of the results . One reviewer wondered why the authors started with the basal calcium experiment? " I like it. I think it is interesting. But it seems slightly unrelated to much of the rest of the paper and is of a very different flavor. It also seems almost unrelated to the introduction. It is not very well integrated into the rest of the paper. It seems as if it would be better to move it to near Figure 7."

We agree with the reviewer that the basal Ca^2+^ experiments could be better introduced and integrated in the manuscript. We would like to start the manuscript with physiological stimuli (action potentials) and proceed with unphysiological UV uncaging for a mechanistic analysis. We therefore favor this order of presentation. However, we now introduced priming and recruitment better, and revised the introduction as mentioned above to better motivate the story flow.

7. Distant mossy fibers on the same axon could complicate capacitance measurements. Have the authors considered how this could affect their capacitance measurements?

This is an interesting point, which was addressed previously at hippocampal mossy fiber boutons (Hallermann et al., PNAS, 2003). In that study, multi-compartment models of the bouton, filopodia and adjacent axons were used to judge how the capacitance measurements could be influenced by neighboring cellular structures. The results indicate that high-frequency sine-wave techniques as used in our study (5 and 10 kHz) probe changes in capacitance of the bouton accurately (<10 % error) and are hardly affected by release from neighboring boutons. Note, that more low frequency signals such as Ca^2+^ currents from neighboring boutons during depolarizations for 1 to 30 ms can indeed contaminate the Ca^2+^ current recordings, in which case the experiments were excluded from the analysis as previously described (Ritzau-Jost et al., Neuron, 2014).

8. How might endocytosis influence the capacitance measurements?

Endocytosis might indeed complicate the measurements of the rates of vesicle fusion. This is an unavoidable error in capacitance measurements in general. For the dose response curve of vesicle fusion, we focused on the initial fast component of release occurring within the first 10 ms, which we assume is less affected by endocytosis, since it is still earlier than the fastest reported form of endocytosis (Watanabe et al., Nature, 2013; Delvendahl et al., Neuron, 2016). However, the sustained release between 10 and 100 ms (Fig. 5) could well be contaminated by endocytosis. We therefore separated the sustained release rate measured with capacitance measurements and deconvolution techniques (now Fig. 5B and C). Both techniques have different limitations and indeed we found some differences between these two techniques (see response to the following comments).

9. On line 245-247 the authors say that they did simultaneous capacitance and deconvolution measurements. Please present these data in a supplementary figure to show the extent to which they agree and disagree. It would be useful to have a quantitative comparison of these two approaches. This is particularly important because it seems as if there is a big difference between the two approaches in 5B.

We thank the reviewer for raising this point. We have added a supplementary figure (Figure 3 – figure supplement 4) showing a correlation between the time constants obtained from capacitance measurements versus the time constants obtained from deconvolution analysis for the recordings where we have both measurements simultaneously. Please note that our results are consistent with a previous comparison of capacitance measurements and this type of deconvolution analysis (Ritzau-Jost et al., Neuron, 2014).

The sustained release rate differs mostly in the used Ca^2+^ concentration for deconvolution and capacitance measurements. However, we thank the reviewer for realizing that there is indeed a difference between these two data sets. The release rate observed with deconvolution analysis is about two times higher compared with a release rate observed with capacitance measurements (cf. Fig. 5B and C). However, the difference is small compared with the large scatter in the data with a 100-fold difference in the rate of release between individual cerebellar mossy fiber boutons. Furthermore, there is a weak but significant correlation in the data set observed with deconvolution analysis compared with the capacitance data. We therefore split the data in 5B into two separate figures based on the method used to estimate the number of vesicles (new Fig. 5B and C). In the manuscript we now carefully discuss the limitations of each of these technique (line 676-681):

“However, it should be mentioned that measuring the sustained release rate is technically very challenging with both presynaptic capacitance and postsynaptic current recordings, because the former cannot differentiate between exo- and endocytosis occurring simultaneously, and the latter can fail to dissect direct release from spill-over current, which is prominent at this synapse (DiGregorio et al., 2002). Therefore, these results must be considered cautiously”.

10. One reviewer raised a concern with Fig 5B. It seems as if the capacitance is measuring from the entire presynaptic bouton (release onto many granule cells) and the granule cell recording is extremely complicated because it has a direct component and a scaled down spillover component that is difficult to convert into vesicle number.

We used the deconvolution technique that was established by Neher and Sakaba (J Neurosci, 2001), which we adjusted for the cerebellar mossy fiber to granule cell synapse (Ritzau-Jost et al., Neuron, 2014). This deconvolution method takes into account the glutamate spill-over and the delayed clearance of the neurotransmitter. Despite our efforts to establish deconvolution techniques and capacitance measurements, we agree that both techniques have different intrinsic limitations when investigating strong long-lasting release. In order to address the reviewer’s concern, we split the data in 5B into two separate figures based on the method used to estimate the number of vesicles released (new Fig. 5B and C). As mentioned above, we now carefully discuss the limitations of these approaches.

11. Please add additional discussion of the direct release and the spillover release components at mossy fiber synapses and how it could affect their measurements.

We have added additional discussion to address this point. Please see response to the above comments.

12. On page 32, it would seem that if you did simulations, you would expect an overshoot. Why isn't there an overshoot for brief uncaging?

There is no overshoot due to the recent improvements in the quantification of Ca^2+^ and Mg^2+^ binding and unbinding kinetics in particular of the photoproducts (Faas et al., Biophys J, 2005; Faas et al., Plos Biology, 2007). In addition, in our simulations ATP and endogenous buffers were taken into account, which further dampened the initial Ca^2+^ spike, as already anticipated in Bollmann et al. (Science, 2000).

13. The paper would great benefit from a more complete Ca2+ dose response curve. Because the authors goal is to completely characterize how release is regulated at the cerebellar mossy fiber bouton, it would significantly strengthen the conclusions of the manuscript to provide data on release rates at Ca2+ concentrations under 1 µM and higher than 50 µM in the dose response curve. We recognize that these experiments may not be completely feasible, but the issue should be openly discussed in the manuscript. The authors demonstrate that the release machinery is not saturated up to 50 µM Ca2+. It would be ideal to determine if the release machinery can be saturated.

We agree with the reviewer that it would be interesting to extend the dose response curve to values below 1 µM. To our knowledge this has only been achieved in a few studies performed at the calyx of Held synapse. As the reviewers know, direct presynaptic recordings and simultaneous pre- and postsynaptic recordings at the cerebellar mossy fiber to granule cell synapse are technically extremely challenging, and the number of experiments in the present paper is already at 139 (80 presynaptic and 59 paired recordings). Therefore, we think that a detailed analysis of the Ca^2+^ dependence of release in the Ca^2+^ range below 1 µM could well form the basis of a separate manuscript. Regarding the upper end of our dose-response curve, we reached the limit of resolving the release rates at 30 to 50 µM Ca^2+^ concentration, although we increased the sampling frequency of our capacitance recordings to a high frequency (10 kHz). Therefore, with the current techniques, it seems impossible to measure the release rate at this synapse at Ca^2+^ levels higher than 50 µM. To explain the limitations of our study we added the following sentences in the discussion:

“Second, we could not investigate allosteric or two-sensor models (Lou et al., 2005; Sun et al., 2007) because we did not address the release rates in the low Ca^2+^ range (<1 µM), therefore, these questions remain to be investigated at the cMFBs. Finally, currently available techniques to estimate fast release rates in the Ca^2+^ range above 50 µM are limited by the sampling frequency of capacitance measurements and dendritic filtering which could prevent the detection of saturation at the upper end of the dose-response curve.”

[Editors' note: further revisions were suggested prior to acceptance, as described below.]

Reviewer #1:This is an impressive paper. Its great strength is that it addresses an important question so carefully and thoroughly that it makes a substantial contribution to the field.My one quibble it that I still think they have done a woefully inadequate job of discussing a great deal of previous literature centered around the residual calcium hypothesis. They have added a new section and an eclectic set of new references (55-80), but they disregard many classic papers and relevant reviews in the field. The residual calcium hypothesis has been most extensively discussed in terms of forms of short-term synaptic plasticity, such as facilitation, augmentation and post-tetanic potentiation. The literature goes about 70 years. I will leave it up to the authors to fix the issue to their satisfaction. Even though it is a minor point, I think this reflects badly on the authors. I leave it up to them to fix it to their own satisfaction.

We thank the reviewer for pointing this out. We fully agree that the manuscript profits from further discussion of our results on vesicle priming in the context of the ‘residual calcium hypothesis’ because these two mechanisms are very likely to converge into a mechanism explaining synaptic facilitation. Therefore, we followed the reviewer’s suggestion and explained in more details the potential interplay between residual calcium, vesicle priming, and synaptic facilitation in a new section in the discussion:

“Implications for synaptic facilitation

Our data might contribute to a better understanding of the mechanisms of the ‘residual Ca^2+^ hypothesis’ explaining synaptic facilitation (Jackman and Regehr, 2017; Katz and Miledi, 1968; Magleby, 1987; Zucker and Regehr, 2002). The strong dependence of the action potential-evoked release on basal Ca^2+^ (Fig. 1) supports the critical effect of residual Ca^2+^ on synaptic strength. Our mechanistic analysis (particularly Fig. 7) indicates that the number of release-ready vesicles rather than the vesicular release probability is regulated by residual Ca^2+^. The high-affinity Ca^2+^ sensor Synaptotagmin-7 (Sugita et al., 2002) could be a sensor for the changes in basal Ca^2+^ levels and mediate the here-reported three-fold increase in synaptic strength (Figs. 1 and 7). Synaptotagmin‑7 has been shown to mediate vesicle recruitment (Liu et al., 2014), asynchronous release (Luo and Südhof, 2017), and synaptic facilitation (Chen et al., 2017; Jackman et al., 2016). If the recruitment and priming steps are fast enough, they could provide a powerful mechanism for synaptic facilitation. Indeed, there is increasing evidence for ultra-fast Ca^2+^-dependent recruitment and priming (reviewed in Neher and Brose 2018) as well as facilitation mediated by an increase in the number of release-ready vesicles rather than the vesicular release probability (Jackman et al., 2016; Kobbersmed et al., 2020; Vevea et al., 2021). Our data are therefore consistent with the emerging view that facilitation is mediated by rapid Synaptotagmin‑7/Ca^2+^-dependent recruitment and priming of vesicles.”

Furthermore, we more precisely defined the aim of this study and introduced the relation with facilitation in the introduction:

“We refer to vesicle priming as the molecular and positional preparation of vesicles for fusion near Ca^2+^ channels (Neher and Sakaba, 2008). According to Imig et al. (2014) molecular priming is the functional correlate of vesicle docking (Imig et al., 2014; Maus et al., 2020). Vesicle replenishment refers to the delivery of new vesicles during sustained activity. The effect of the residual Ca^2+^ on the strength of synapses particularly during synaptic facilitation has been studied for decades with a particular focus on the release probability of vesicles (see discussion). Here we investigate the Ca^2+^-dependence of priming and replenishment, which increases the number of release-ready vesicles.”

Reviewer #2:The authors have done a good job responding to the reviews. Their response to point 7 should have led to a comment (about the complications of distal terminals to Cm measurements) somewhere in the methods, rather than just a comment to the reviewer. Otherwise, I have no further concerns with the work and feel it is ready for publication. It is a fine paper.

We followed the reviewer’s suggestion and included this in the methods section:

“Distant mossy fiber boutons on the same axon are unlikely to contaminate capacitance measurements because investigations at hippocampal mossy fiber buttons indicate that the high-frequency sine-wave techniques as used in our study are hardly affected by release from neighboring boutons on the same axon (Hallermann et al., 2003).”